

# Modeling the effect of non-ideality, dynamic mass transfer and viscosity on SOA formation in a 3-D air quality model

Youngseob Kim[1], Karine Sartelet[1], and Florian Couvidat[2]

[1]CEREA, Joint Laboratory École des Ponts ParisTech / EDF R&D, Université Paris-Est, 77455 Champs-sur-Marne, France
[2]Institut National de l'Environnement Industriel et des Risques, 60550 Verneuil-en-Halatte, France

*Correspondence to:* Y. Kim (youngseob.kim@enpc.fr)

**Abstract.**

In this study, common assumptions (ideality and thermodynamic equilibrium) commonly made in 3-dimensional (3-D) air quality models were reconsidered to evaluate their impacts on secondary organic aerosol (SOA) formation over Europe.

To investigate the effects of non-ideality, dynamic mass transfer and aerosol viscosity on the SOA formation, the Secondary
Organic Aerosol Processor (SOAP) model was implemented in the 3-D air quality model Polyphemus. This study presents the first 3-D modeling simulation which describes the impact of aerosol viscosity on the SOA formation. The model uses either the equilibrium approach or the dynamic approach with a method specially designed for 3-D air quality models to solve efficiently particle-phase diffusion when particles are viscous.

Sensitivity simulations using two organic aerosol models implemented in Polyphemus to represent mass transfer between
gas and particle phases show that the computation of the absorbing aerosol mass strongly influences the SOA formation. In particular, taking into account the concentrations of inorganic aerosols and hydrophilic organic aerosols in the absorbing mass of the aqueous phase increases the average SOA concentration by 5% and 6%, respectively. However, inorganic aerosols influence the SOA formation not only because they constitute an absorbing mass for hydrophilic SOA, but also because they interact with organic compounds. Non-ideality (short, medium and long-range interactions) was found to influence SOA concentrations by
about 30%.

Concerning the dynamic mass transfer for the SOA formation, if the viscosity of SOA is not taken into account and if ideality of aerosols is assumed, the dynamic approach is found to give generally similar results than the equilibrium approach (indicating that equilibrium is an efficient hypothesis for inviscid and ideal aerosols). However, when a non-ideal aerosol is assumed, taking into account the dynamic mass transfer leads to a decrease of concentrations of the hydrophilic compounds
(compared to equilibrium). This decrease is due to differences in the values of activity coefficients, which are different between values computed for bulk aerosols and those for each size section. This result indicates the importance of non-ideality on the dynamic evolution of SOA.

For viscous aerosols, assuming a highly viscous organic phase leads to an increase in SOA concentrations during daytime (by preventing the evaporation of the most volatile organic compounds). The partitioning of non-volatile compounds is not
affected by viscosity, but the aging of more volatile compounds (that leads to the formation of the less volatile compounds) slows down as the evaporation of those compounds is stopped due to the viscosity of the particle. These results imply that



aerosol concentrations may deviate significantly from equilibrium as the gas-particle partitioning could be higher than predicted by equilibrium. Furthermore, although a compound evaporates in the simulation using the equilibrium approach, the same compound can condense in the simulation using the dynamic approach if the particles are viscous.

The results of this study emphasize the need for 3D air quality models to take into account the effect of non-ideality on SOA formation and the effect of aerosol viscosity for the more volatile fraction of semi-volatile organic compounds.

## 1   Introduction

Inorganic and organic aerosols constitute an important fraction of aerosols (Putaud et al., 2004; Jimenez et al., 2009), which influence climate and health (Rattanavaraha et al., 2011; Boucher, 2015; Shrivastava et al., 2017). A large fraction of inorganic and organic aerosols are not directly emitted, but they are formed in the atmosphere by the condensation of condensable compounds, which are often semi volatile, i.e. they exist both in the gas and in the particle phases. The modeling of the mass transfer of condensable compounds (inorganic and organic) from the gas phase to the particle phase is important, because it determines the fraction of the condensable compounds in the particle phase, and therefore the particle concentration. It is usually modeled by three approaches: the dynamic approach, the equilibrium approach, and the hybrid approach. In the dynamic approach, the mass transfer between the gas and the particle phases is explicitly calculated by solving the mass flux equation (e.g., Wexler and Seinfeld, 1990; Bowman et al., 1997). In the equilibrium approach, instantaneous equilibrium is assumed between the gas and particle phases (Pankow, 1994). The dynamic approach provides a more accurate representation of the gas/particle mass transfer but is computationally more expensive than the equilibrium approach. The hybrid approach combines these two approaches. As gas-phase molecules condense more rapidly on fine than on coarse particles (and therefore reach more rapidly equilibrium with fine particles), the mass transfer is explicitly computed for coarse particles using the dynamic approach and instantaneous equilibrium is assumed for fine particles in the hybrid approach (Capaldo et al., 2000; Zhang et al., 2004; Debry et al., 2007).

Several previous studies showed that organic aerosols can be highly viscous (Virtanen et al., 2010; Cappa and Wilson, 2011; Pfrang et al., 2011; Shiraiwa et al., 2011; Vaden et al., 2011; Shiraiwa and Seinfeld, 2012; Abramson et al., 2013; Renbaum-Wolff et al., 2013; Shiraiwa et al., 2013). The diffusion of organic compounds from the particle surface inside the particle is influenced by the viscosity of organic aerosols, which depends on relative humidity and aerosol composition (Song et al., 2016; O'Meara et al., 2017). The diffusion is very slow when the particle phase state is semi-solid, solid or glassy solid (Shiraiwa et al., 2017). By influencing the diffusion inside the particle, viscosity influences the mass transfer between gas and particle phases, which is much slower than for non-viscous particles (Shrivastava et al., 2017). Several models explicitly treat the particle-phase diffusion of organic compounds (Shiraiwa et al., 2012; Roldin et al., 2014). However, the use of these models is limited in 3-dimensional (3-D) air quality models because particles need to be discretized with a high number of particle layers, which leads to an expensive computational cost.

Although the equilibrium approach is widely used in 3-D air quality models because of its computational efficiency, the dynamic approach is also sometimes used in 3-D air quality models for inorganic aerosols (Jacobson, 1997; Meng et al., 1998;



Sun and Wexler, 1998; Sartelet et al., 2007a). However, to our knowledge, the impact of viscosity of particles on gas/particle phase partitioning and organic aerosol concentrations has yet not been taken into account in 3-D air quality models.

The mass transfer of condensable organic compounds between the gas and particle phases is influenced by interactions with other compounds. The activity coefficients reflect the non-ideality of aerosols and the influence of the interactions between compounds on the mass transfer between the gas and particle phases.

Organic aerosol models often assume ideality, and they do not take into account the influence of activity coefficients on the formation of secondary organic aerosols. However, activity coefficients may be determined by thermodynamic models. For example, the UNIversal Functional group Activity Coefficient (UNIFAC) thermodynamic model (Fredenslund et al., 1975) is based on a functional group method, which estimates short-range activity coefficients (interactions between uncharged molecules) by using the structure of the molecules present in the particles. However, in the aqueous phase, for hydrophilic organic compounds, due to the presence of ions, such as inorganic ions, medium and long-range activity coefficients (resulting from electrostatic interactions) may also influence activity coefficients. These medium and long-range activity coefficients are described by the Aerosol Inorganic–Organic Mixtures Functional groups Activity Coefficient AIOMFAC model (Zuend et al., 2008). The effect of activity coefficients was already investigated by a previous study (Couvidat et al., 2012) by using the UNIFAC model. Compared to assuming ideality, computing activity coefficients was found to decrease the concentrations of hydrophobic SOA (condensing onto the organic phase of particles) but also to increase the concentrations of hydrophilic SOA (condensing onto the aqueous phase of particles). Pye et al. (2017) obtained a reduction of the bias in SOA for routine monitoring stations taking into account the non-ideality via activity coefficients. However, this study did not take into account the effect of interactions between inorganic and organic compounds.

The Secondary Organic Aerosol Processor (SOAP) model (Couvidat and Sartelet, 2015) has been developed to represent the condensation/evaporation of organic aerosols using both the equilibrium and dynamic approaches. The SOAP model was designed to be implemented in 3-D air quality models and can be used to implicitly represent the diffusion of organic compounds inside the particle phase, using a low number of particle layers. Compared to an explicit representation of the diffusion of organic compounds with a high number of particle layers, the SOAP model showed good agreements of modeled organic concentrations of viscous particles, using a lower number of aerosol layers.

In this study, we present the implementation of the SOAP model in the 3-D air quality model Polyphemus, and present differences between SOAP and the hydrophilic/hydrophobic organic ($H^2O$) model (Couvidat et al., 2012) and how the new processes implemented in the SOAP model influence SOA formation (absorbing mass, non-ideality, viscosity). Jathar et al. (2016) showed that organic-phase water uptake leads to an increase in total organic aerosol concentration. Water uptake is taken into account in the SOAP model to estimate the absorbing mass. We estimate for the first time in a 3D air quality model the maximum influence of aerosol viscosity on particle organic concentrations over Europe. To do so, we compare simulations assuming inviscid or extremely viscous aerosols. We also estimate the influence of non-ideality, in particular the influence of inorganic concentrations via medium and long-range activity coefficients on SOA concentrations. The SOAP model and differences with $H^2O$, the previously used SOA model in Polyphemus, are described in section 2. Section 3 details the modeling



of the newly added processes studied here: medium and long-range activity coefficients, aerosol dynamic, viscosity. Finally, section 4 presents the simulations, and the sensitivity to these processes.

## 2 Description and implementation of SOAP in Polyphemus

The SOAP model was implemented in the chemistry transport model Polair3D (Sartelet et al., 2007b) of the air quality platform Polyphemus version 1.8 (Mallet et al., 2007). The aerosol dynamics (coagulation, condensation/evaporation) is modeled with the SIze REsolved Aerosol Model (SIREAM, Debry et al. (2007)). The particle size distribution is divided into sections, each section corresponding to a range of diameters. Similarly to H$^2$O, the SOAP model is based on the molecular surrogate approach. It distinguishes hydrophobic compounds (condensing into the organic phase of particles) from hydrophilic compounds (condensing into the aqueous phase of particles).

In the molecular surrogate approach, organic compounds are represented by surrogates, which are model compounds chosen depending on their sources (anthropogenic vs biogenic) and their properties, such as their affinity with water (hydrophilic vs hydrophobic) and their volatility. Oxidation of the SOA precursors differs depending on the regime of nitrogen oxides (NO$_x$) (low NO$_x$ regime vs high NO$_x$ regime). Different reactions (Kim et al., 2011) were added to the gas-phase chemistry model of Polyphemus (CB05 is used here, Yarwood et al. (2005)) to model the formation of organic compounds from five classes of SOA precursors (intermediate and semi-volatile organic compounds of anthropogenic emissions, aromatic compounds, isoprene, monoterpenes and sesquiterpenes). As detailed in Couvidat et al. (2012), surrogates from anthropogenic precursors are mostly hydrophobic, while those from biogenic precursors are mostly hydrophilic. Table 1 summarizes the surrogates and their properties (volatility, hydrophilicity).

In previous studies using Polyphemus (Couvidat et al., 2012; Sartelet et al., 2012; Couvidat et al., 2013; Zhu et al., 2016), the H$^2$O model was used to partition organics between the gas and particle phases: instantaneous equilibrium was assumed between the gas and particle phases, and only short-range activity coefficients were taken into account. They were computed with the UNIQUAC Functional-group Activity Coefficients (UNIFAC) model (Fredenslund et al., 1975), i.e. without taking into account the impact of inorganic compounds, as if the aqueous phase is only constituted of water and organics. In other words, H$^2$O only takes into account solvents into the computation of short range interactions and H$^2$O implicitly assumed that organics have no interaction with inorganics.

However, AIOMFAC (Zuend et al., 2008) is a thermodynamic model designed for the calculation of activity coefficients of different chemical species in inorganic-organic mixtures. It takes into account the short-range, middle-range and long-range interactions between molecules and ions. For short-range interactions, AIOMFAC differs from UNIFAC because it takes inorganics into account in short-range interactions by taking relative van der Waals subgroup volume and surface area UNIFAC parameters. It assumes that interaction parameters of inorganics with organics for short-range interactions are zero: the short range organic-inorganic interactions are ideal.

The SOAP model inherits all the characteristics of the H$^2$O model and new processes (such as modeling of inorganic-organic interactions via activity coefficients, and dynamic evolution of gas/particle partitioning) are added (Couvidat and



**Table 1.** Description of the SOA surrogate compounds (Couvidat and Sartelet, 2015).

| Surrogate | Type* | Precursors | Conditions of formation | Volatility† |
|---|---|---|---|---|
| BiMT | A | isoprene | Oxidation by OH (low $NO_x$) | high |
| BiPER | A | isoprene | Oxidation by OH (low $NO_x$) | high |
| BiDER | A | isoprene | Oxidation by OH (low $NO_x$) | medium |
| BiMGA | A | isoprene | Oxidation by OH (high $NO_x$) | medium |
| BiNGA | B | isoprene | Oxidation by OH (high $NO_x$) | high |
| BiNIT3 | B | isoprene | Oxidation by $NO_3$ | high |
| BiA0D | A | monoterpenes | Oxidation by OH and $O_3$ | very low if the qaueous aerosol is acidic |
| BiA1D | A | monoterpenes | Oxidation by OH and $O_3$ | medium |
| BiA2D | A | monoterpenes | Oxidation by OH and $O_3$ | medium |
| BiNIT | B | monoterpenes | Oxidation by $NO_3$ | high |
| BiBlP | B | sesquiterpenes | Oxidation by OH | very low |
| BiBmP | B | sesquiterpenes | Oxidation by OH | medium |
| AnBlP | B | aromatics | Oxidation by OH (low $NO_x$) | low |
| AnBmP | B | aromatics | Oxidation by OH (low $NO_x$) | high |
| AnClP | B | aromatics | Oxidation by OH (high $NO_x$) | non-volatile |
| POAlP | B | - | Primary SVOC | low |
| POAmP | B | - | Primary SVOC | high |
| POAhP | B | - | Primary SVOC | very high |
| SOAlP | B | POAlP | Oxidation by OH | very low |
| SOAmP | B | POAmP | Oxidation by OH | low |
| SOAhP | B | POAhP | Oxidation by OH | high |

*: A and B correspond to hydrophilic and hydrophobic compounds, respectively.

†: very low for compounds with $K_p > 100$, low for compounds with $100 \geq K_p > 1$, medium for compounds with $1 \geq K_p > 0.1$, high for compounds with $0.1 \geq K_p > 0.01$ and very high for compounds with $K_p \leq 0.01$.

Sartelet, 2015). However, SOAP differs from $H^2O$ not only because of the possibility to model inorganic-organic interactions via activity coefficients, and to model dynamically the gas/particle partitioning of viscous aerosols, but also differences occur in the computation of the gas/particle partitioning due to the computation of the absorbing mass.

In SOAP and $H^2O$, ideality is defined by reference to the pure state for hydrophobic compounds (activity coefficients are equal to one when the compound is pure) and to the infinite dilution state for hydrophilic compounds (activity coefficients are equal to one when the compounds is diluted into an infinite amount of water). The partitioning is computed according to Raoult's law for hydrophobic compounds and to Henry's law for hydrophilic compounds.





The differences between SOAP and H$^2$O are now detailed and their impact on previously published simulations of Polyphemus/H$^2$O is quantitatively assessed.

## 2.1 Composition of aerosols in the aqueous and organic phases

The equilibrium approach is used in the H$^2$O model, and it can be used in the SOAP model. In this approach, the partitioning between the gas and particle organic phases is done following Pankow (1994):

$$\frac{c_{p,i}}{c_{g,i}} = K_{p,i}\, c_p \tag{1}$$

where $K_{p,i}$ is the organic-phase gas/particle partitioning coefficient (m$^3$/µg), $c_{p,i}$ is concentration of the compounds $i$ in the organic phase, $c_{g,i}$ is the gas-phase concentration, $c_p$ (µg/m$^3$) is the total concentration of the particles in the organic phase.

Whereas in the H$^2$O model, $c_p$ is only the concentration of the organic compounds in the particles, in the SOAP model the absorption of water by the organic phase, $c_{water,p}$ (µg/m$^3$) , is also included in $c_p$.

The absorption of water by the organic phase is computed using Equation 2 following Couvidat and Sartelet (2015).

$$c_{water,p} = \frac{c_p\, M_{water}\, RH}{\gamma_{water,p}\, M_p} \tag{2}$$

where $M_{water}$ is the molar mass of water (g/mol), RH is the relative humidity, $\gamma_{water,p}$ is the activity coefficient of water in

the organic phase and $M_p$ is the averaged molar mass of the organic phase (g/mol).

The partitioning between the gas and the aqueous phases is done similarly as in the organic phase:

$$\frac{c_{aq,i}}{c_{g,i}} = K_{aq,i}\, c_{aq} \tag{3}$$

where $c_{aq,i}$ is the aqueous-phase concentration of the compound $i$ (µg/m$^3$), $K_{aq,i}$ is the aqueous-phase gas/particle partitioning coefficient (m$^3$/µg), and $c_{aq}$ (µg/m$^3$) is the total concentration of the particles in the aqueous phase. $K_{aq,i}$ is computed as

detailed in Couvidat and Sartelet (2015) and depends on the activity coefficient. In the H$^2$O model, $c_{aq}$ corresponds only to the liquid water content (LWC) calculated using a thermodynamic model, e.g., ISORROPIA (Nenes et al., 1999), for inorganic aerosols. However, $c_{aq}$ includes inorganic aerosols, hydrophilic organic aerosols, and absorbed water by hydrophilic organic aerosols in addition to LWC in the SOAP model. The larger concentrations of $c_{aq}$ in the SOAP model than in the H$^2$O model lead in return to larger compounds concentrations in the aqueous phase ($c_{aq,i}$).

## 2.2 Impact on SOA concentrations

Sensitivity simulations are performed to quantify the impact on organic concentrations of the differences between the H$^2$O and SOAP models in the formulation of the absorbing mass ($c_p$ and $c_{aq}$) used in the modeling (the total particle concentrations of the organic and aqueous phases respectively). Within the Polyphemus platform, the two SOA models are implemented with



the Size Resolved Aerosol Model (SIREAM) aerosol module (Debry et al., 2007). The simulations of Couvidat et al. (2012)
are rerun using the SOAP model instead of $H_2O$. The model configuration is detailed in Couvidat et al. (2012). The simulation
domain covers Europe (see Figure 1) with a horizontal resolution of $0.5° \times 0.5°$ and 9 vertical levels (20 m, 80 m, 210 m,
550 m, 1150 m, 1950 m, 2950 m, 4750 m, 9000 m). The initial and boundary conditions are calculated using data from global

**Table 2.** List of the sensitivity simulations to compare the $H_2O$ and SOAP models.

| Simulation name | SOA model | Aqueous-phase particle included in $c_{aq}^{\dagger}$ | Organic-phase particle included in $c_p^{\dagger}$ | Activity coefficient |
|---|---|---|---|---|
| SOAP-sr | SOAP | - Water absorbed by inorganic aerosol<br>- Inorganic aerosol<br>- Hydrophilic organic aerosol<br>- Water absorbed by hydrophilic organic aerosol | - Hydrophobic organic aerosol<br>- Water absorbed by hydrophobic organic aerosol | UNIFAC-sr** |
| SOAP-Reference | SOAP | - Water absorbed by inorganic aerosol<br>- Inorganic aerosol<br>- Hydrophilic organic aerosol<br>- Water absorbed by hydrophilic organic aerosol | - Hydrophobic organic aerosol<br>- Water absorbed by hydrophobic organic aerosol | UNIFAC |
| SOAP-no_inorg | SOAP | - Water absorbed by inorganic aerosol<br>- Hydrophilic organic aerosol<br>- Water absorbed by hydrophilic organic aerosol | - Hydrophobic organic aerosol<br>- Water absorbed by hydrophobic organic aerosol | UNIFAC |
| SOAP-no_water | SOAP | - Water absorbed by inorganic aerosol<br>- Hydrophilic organic aerosol | - Hydrophobic organic aerosol | UNIFAC |
| SOAP-ideal | SOAP | - Water absorbed by inorganic aerosol | - Hydrophobic organic aerosol | ideal* |
| $H_2O$-Reference | $H_2O$ | - Water absorbed by inorganic aerosol | - Hydrophobic organic aerosol | UNIFAC |
| SOAP-basic | SOAP | | | |
| $H_2O$-Ideal | $H_2O$ | - Water absorbed by inorganic aerosol | - Hydrophobic organic aerosol | ideal* |

$\dagger$: total particle concentration of the organic phase ($c_p$) and aqueous phase ($c_{aq}$).

*: ideal mixture, activity coefficient is set to 1.0.

**: short range activity coefficients are calculated taking into account inorganic aerosols.





models MOZART (gas) and ECHAM5-HAMMOZ (particles). Anthropogenic emissions are taken from the EMEP (European Monitoring and Evaluation Programme) inventory (http://www.ceip.at/) and biogenic emissions are estimated with MEGAN (Model of Emissions of Gases and Aerosols from Nature) (Guenther et al., 2006).

Sensitivity simulations are conducted for June 2002. The sensitivity tests are detailed in Table 2. Domain-averaged concentrations of SOA are used to compare the sensitivity simulations in Figure 2. In the SOAP-Reference simulation, $c_{aq}$ is computed by considering the water absorbed by inorganic aerosols and by hydrophilic aerosols (inorganic aerosols and hydrophilic organic aerosols) while $c_p$ is computed by considering hydrophobic organic aerosols and the water absorbed by hydrophobic organic aerosols. Overall, the average difference in SOA concentrations between the SOAP-Reference and $H^2O$-Reference simulations is 15%. The differences between these two simulations are mostly due to the influence of the different compounds included in the absorbing mass used for the partitioning of the gas and particle phases, i.e. in the computation of $c_p$ and $c_{aq}$ (the total particle concentrations of the organic and aqueous phases). Simulations using the same absorbing mass in SOAP and $H^2O$ (water absorbed by inorganic aerosols for $c_{aq}$ and hydrophobic organic aerosol for $c_p$) lead to similar concentrations (see the comparison of the simulations $H^2O$-ideal and SOAP-ideal in Figure 2). Adding water absorbed by organic aerosols in the absorbing mass leads to a slight increase in SOA concentration (comparisons of the simulations SOAP-no_inorg and SOAP-no_water). Adding inorganic aerosols in the absorbing mass of hydrophilic aerosols ($c_{aq}$) has a larger impact than the addition of water absorbed by organic aerosols (5%, see the comparison of the simulations SOAP-Reference and SOA-no_inorg). Adding organic aerosols in the absorbing mass of $c_{aq}$ has an impact as large as inorganic aerosols (6%, see the comparison of the simulations SOAP-no_water and SOAP-basic, which is similar to $H^2O$-Reference).

Not only the absorbing mass strongly influences the SOA concentrations, but also the interactions between compounds, as modeled by activity coefficients. The influence of taking into account organic-organic interactions by short-range activity coefficients is as high as 18% (see the comparison between the simulations $H^2O$-Reference and $H^2O$-Ideal). The difference between the SOAP-Reference and SOAP-Ideal simulations is much larger (35%), because of differences in the computation of the absorbing mass between SOAP and $H^2O$.

An additional sensitivity simulation SOAP-sr is used to estimate UNIFAC sensitivities when inorganic aerosols are added in the computation of short-range activity coefficient as in AIOMFAC. The averaged SOA concentrations in SOAP-sr increase by 15% compared to those of SOAP-Reference. This difference is further discussed in section 4.3. As inorganic-organic interaction parameters are set to zero in UNIFAC, taking into account inorganics in the computation of short-range activity coefficients (simulation SOAP-sr) leads to activity coefficients closer to the pure compound state and therefore to a decrease of activity coefficients (as organics are generally more stable at pure state than in water). As activity coefficients are lower in SOAP-sr than in SOAP-Reference, organic concentrations are higher.

Figure 3 shows the horizontal distribution of (a) SOA concentrations obtained by the SOAP-Reference simulation, and (b) the differences between the SOAP-Reference and $H^2O$-Reference simulations. As expected, the SOA concentrations are higher in the SOAP-Reference simulation than in the $H^2O$-Reference simulation. Depending on the location, the differences in the SOA concentrations between the simulations are due to different compounds used to compute the partitioning between the gas and particle phases. Over North-Eastern Europe, the differences are due to the large hydrophilic biogenic organic aerosol





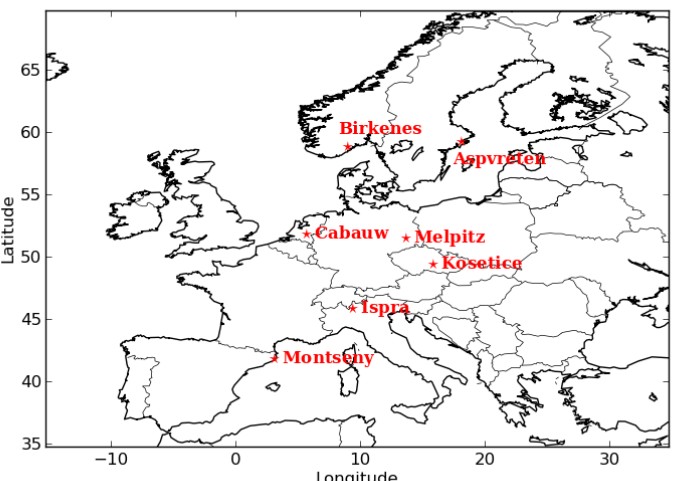

**Figure 1.** Simulation domain and location of observation stations.

concentrations. Taking them into account in the computation of $c_{aq}$ strongly increases the concentrations of monoterpenes SOA over South-Western Europe (especially in Northern Italy where simulated concentrations of nitrate are high). Over northern Italy, large differences are also observed for anthropogenic aromatic organic aerosol concentrations. Even though these

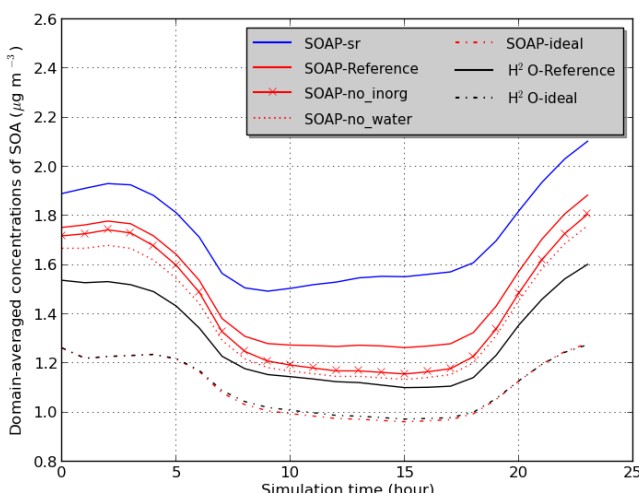

**Figure 2.** Temporal variation of the average SOA concentrations over the domain (see Table 2 for the description of simulations).



compounds are hydrophobic, taking into account the water they absorbed when computing $c_p$ leads to an increase in their con-
centrations. Similarly, near North Africa and the Iberian Peninsula, the concentrations of hydrophilic surrogates from isoprene
oxidation are higher with SOAP than with $H^2O$, because of the large concentrations of sulfate from shipping emissions. Taking
sulfate into account (but without taking into account its influence on activity coefficients) when computing the partitioning
between the gas and particle phases leads to an increase in the concentrations of hydrophilic organic concentrations in the
particle phase.

## 3   Description of the newly added processes

### 3.1   Interaction of inorganic/organic aerosols using the AIOMFAC model

Although activity coefficients are computed with the UNIFAC model for $H^2O$, depending on the user's choice, in the SOAP
model, activity coefficients can be calculated using the UNIFAC or the AIOMFAC model. UNIFAC was developed to re-
produce the interactions between water and organic compounds, which are dominant for a non-electrolyte liquid mixture.
In UNIFAC, organic compounds are represented by different functional groups including alkane, aromatic carbon, alcohol,
carbonyl. Interaction coefficients between water and these functional groups are calculated.

However, for an electrolyte liquid mixture, the mixed organic and inorganic system may influence activity coefficients, by
middle-range and long-range interactions. This influence of inorganic aerosols on the calculation of activity coefficients in the
SOAP model can be estimated by the AIOMFAC model that considers this mixed organic/inorganic system.

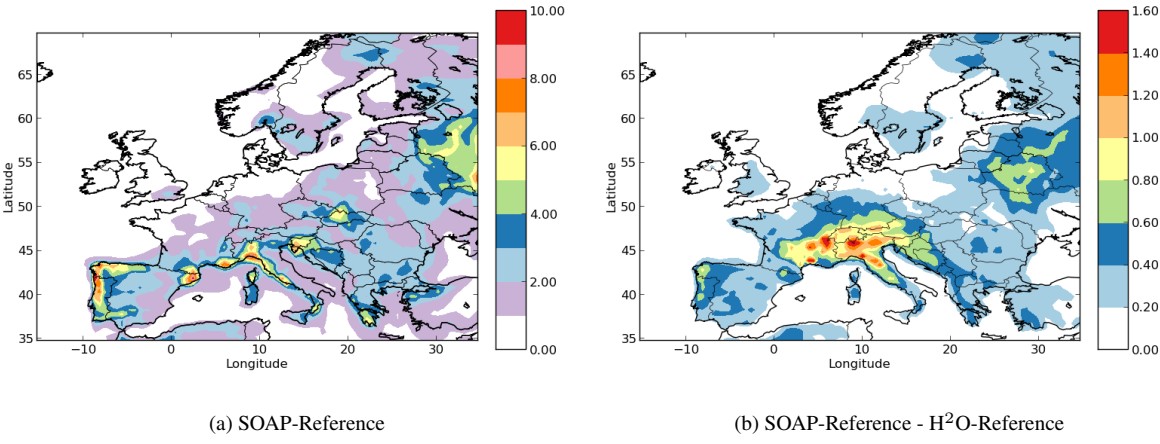

(a) SOAP-Reference                    (b) SOAP-Reference - $H^2O$-Reference

**Figure 3.** SOA concentrations in (a) the SOAP-Reference simulation ($\mu g\ m^{-3}$), (b) the differences between the SOAP-Reference and $H^2O$-
Reference simulations ($\mu g\ m^{-3}$).





The activity coefficient in the AIOMFAC model is calculated by the following equation:

$$\gamma = \gamma_{\text{LR}} \gamma_{\text{MR}} \gamma_{\text{SR}} \tag{4}$$

where $\gamma_{\text{LR}}, \gamma_{\text{MR}}, \gamma_{\text{SR}}$ are the contributions of long-range interactions (electrostatic force between ions), middle-range interactions (interactions between ions and molecular dipoles), and short-range interactions (group-contribution method as in UNIFAC).

### 3.2   Equilibrium and dynamic approaches

Typically, 3-D air quality models mostly use an equilibrium approach to represent condensation/evaporation of aerosols. How-
ever, using a dynamic approach may be necessary if the kinetic effects are large (for example if the diffusion in the organic phase is low due to the high particle viscosity or if condensation over coarse particles occurs). In the SOAP model, depending on the user's choice, either the equilibrium approach or the dynamic approach can be used to model condensation/evaporation. An explicit representation of diffusion inside particles, which would involve discretizing the particle along the radius of the particle, can not be used in 3D air quality models, due to the heavy computation time of such a method. To solve this issue,
a method was developed by Couvidat and Sartelet (2015) to implicitly represent the condensation/evaporation/diffusion of organic compounds for a specified organic phase diffusion coefficient. This method separates the particle into a low number of layers that represent different areas of the particle (the gas/particle interface, the core of the particle and intermediate layers).

    To use the dynamic approach in this study, several simplifications are carried out for hydrophilic compounds. As a dynamic approach is not used to simulate the formation of inorganic aerosols, the thermodynamic model ISORROPIA (Nenes et al.,
1998) with the equilibrium approach is used to estimate the partitioning of inorganics, the aerosol liquid water content and the pH. The pH given by ISORROPIA is used for each size section and the liquid water content is redistributed over sections proportionally to the amount of inorganics.

    In the dynamic approach, the mass transfer rate, $J$ (µg/m$^3$/s) by condensation/evaporation at the gas/particle interface is calculated as follows:

$$J_{\text{cond/evap}} = k_{\text{absorption}} \left( c_g - c_{eq} \right) \tag{5}$$

where $k_{\text{absorption}}$ is the kinetic rate of absorption ($s^{-1}$), $c_g$ is the gas-phase concentration (µg/m$^3$) and $c_{eq}$ is the gas-phase concentration at the interface of particles (µg/m$^3$).

    The kinetic rate of absorption $k_{\text{absorption}}$ is defined as follows (Seinfeld and Pandis, 1998):

$$k_{\text{absorption}} = 2\pi \, d_p \, D_{\text{air}} \, N \, f \left( Kn, \alpha \right) \tag{6}$$

where $d_p$ is the particle mean diameter (m), $D_{\text{air}}$ is the diffusivity of the condensing compounds in air (m$^2$/s), and $N$ is the number concentration of particles (#/m$^3$). The function $f\left(Kn, \alpha\right)$ depends on the Knudsen number ($Kn = \frac{2\lambda}{d_p}$), which is calculated using the mean free path in air $\lambda$ (m), and the accommodation coefficient $\alpha$, which accounts for imperfect surface accommodation. It is taken equal to 0.5 following Saleh et al. (2013).

    For viscous compounds, the condensation/evaporation is limited by the diffusion flux in the internal layers of the particles.





We assume that in each particle layer the evolution of concentration $c_{p,i}$ of species $i$ can be described as a deviation of an equilibrium concentration ($c_{g,i}K_{p,i}c_p$) when the condensation/evaporation of the species is limited by the diffusion of organic compounds in the organic phase.

This deviation can be described by taking into account the flux of diffusion with the mass transfer rate by condensation/evaporation (Equation 36 of Couvidat and Sartelet (2015)).

$$J_{\text{diff}} = k_{\text{diff}}(c_{g,i}K_{p,i}c_p - c_{p,i}) \tag{7}$$

The kinetic rate of diffusion $k_{\text{diff}}$ ($s^{-1}$) is computed as follows (Couvidat and Sartelet, 2015):

$$k_{\text{diff}} \propto \frac{1}{\tau_{\text{diff}}} \tag{8}$$

$\tau_{\text{diff}}$ is the characteristic time (s) for diffusion in the particle:

$$\tau_{\text{diff}} = \frac{R_p^2}{\pi^2 D_{\text{org}}} \tag{9}$$

10   where $R_p$ is the radius of the particle (m) and $D_{\text{org}}$ is the organic-phase diffusivity ($m^2$/s).

The final mass flux by the mixed phenomenon condensation/evaporation/diffusion is computed by assuming that the characteristic time of the combined effect of condensation/evaporation and diffusion is equal to the sum of the characteristic time of condensation/evaporation ($J_{\text{cond/evap}}$) and diffusion ($J_{\text{diff}}$) as follows:

$$\frac{1}{J_{\text{tot}}} = \frac{1}{J_{\text{cond/evap}}} + \frac{1}{J_{\text{diff}}} \tag{10}$$

15   More details on the model may be obtained in Couvidat and Sartelet (2015).

### 3.3   Particle-phase diffusion cases and impact of viscosity on SOA formation

To assess the maximum impact of viscosity on SOA formation: two theoretical studies are studied. The first case, referred hereafter as the "Dynamic inviscid" simulation, assumes that particles are inviscid, i.e. SOA formation is not limited by the particle-phase diffusion: the particle-phase diffusion is so fast that there is no difference in concentrations inside the particle. 20   In this case, compounds condense or evaporate until reaching equilibrium over the whole particle.

The other case, referred as the "Dynamic viscous" simulation, assumes that the particle is "Infinitely viscous" (i.e. too viscous for diffusion to occur inside the particle even at high relative humidity). A very low diffusivity of $10^{-30}$ $m^2$ $s^{-1}$ is assumed. Practically, for simplification purposes, two aerosol layers (the interface and an internal layer) are used in the "Dynamic viscous" simulation. The internal layer and the interface represent 99% and 1% of the aerosol mass, respectively 25   following the method of Couvidat and Sartelet (2015) in which condensation at interface is not limited by particle-phase diffusion. The mass transfer between the gas and particle phases can be described by two steps: the first step between the gas phase and the interface and the second step between the interface and the internal layer. The transfer between layers can happen





without diffusion to assure that the mass fraction of layers (mass of the layer over the mass of the particle) remain constant even in the case of a change in the mass of the particle (growth or shrinking of the particle).

30    The SOA formation for a highly viscous particle is complex. The evolution of the concentration of an organic compound depends on the volatility of the compound with respect to the other compounds. The SOA formation for a highly viscous particle is schematized in Figure 4 in the case of the growth and the shrinking of an extremely viscous particle. In the case of an organic particle growth (mass increase), the condensation of a low-volatility organic compound **B** (its behavior is described by the red curved arrows in Figure 4) onto a growing particle can favor the condensation of a compound **A** of higher volatility (its behavior is described by blue curved arrows in Figure 4) at the interface of the particle (even if the total concentration of **A** inside the particle exceeds equilibrium). The compound **A** condenses onto the new layer created by the compound **B**

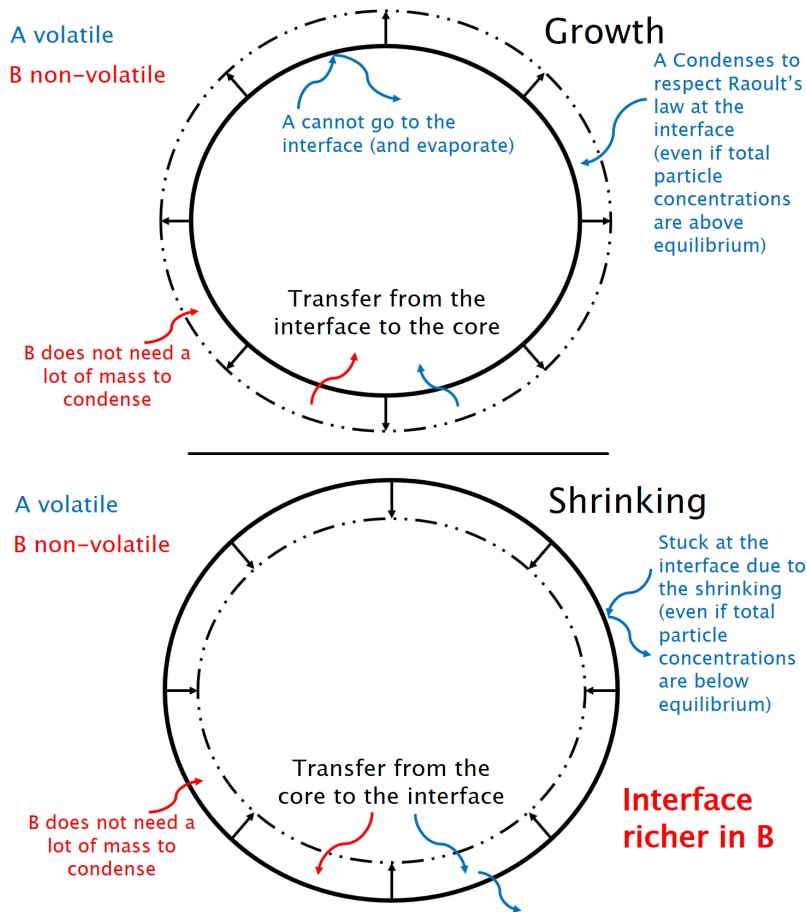

**Figure 4.** Schematic representation of SOA formation for a growing (top) and shrinking (down) highly viscous aerosol. The blue curved arrows describe the behavior of an organic compound **A** and the red curved arrows describe the behavior of a low-volatility organic compound **B**.



to respect Raoult's law at the interface. Moreover, the condensation of the compound **B** and the growth of the particle can prevent evaporation of the compound **A** stuck at the core of the particle (because of the absence of diffusion) and can lead to its "entrapment".

In the case of an organic particle shrinking (mass decrease), a compound **A** inside the particle can evaporate as long as it can reach the interface. The presence of a low-volatility compound **B** at the interface can slow down the evaporation of the compound **A** due to the enrichment of the interface in the low-volatility compound **B**. Moreover condensation of the semi-volatile compound **A** can be strongly slowed down as the compound condensing at the interface is not transferred from the interface to the core of the particle for extremely viscous particles (transfer occurs from the core to the interface to maintain constant mass ratios of the layers).

## 4 Impact on SOA formation

### 4.1 Simulation setup

The chemistry transport model Polair3D of the air-quality modeling platform Polyphemus coupled to the SOAP model is evaluated during summer 2012. The modeling domain covers Europe with a horizontal resolution of $0.5° \times 0.5°$ (see Figure 1). Anthropogenic emissions are generated with the EMEP inventory for 2012. Intermediate and semi-volatile organic compounds (I/S-VOC) emissions are estimated as detailed in Couvidat et al. (2012), by multiplying the primary organic aerosol emissions by a factor 4 and by assigning them to compounds of different volatilities (POAlP, POAmP, POAhP). Biogenic emissions are generated with the MEGAN model (Guenther et al., 2006). ECMWF (European Centre for Medium-Range Weather Forecasts) meteorological reanalysis data (http://www.ecmwf.int/, ERA Interim) are used to calculate meteorological fields. Initial and boundary conditions are obtained from the simulation data of MOZART-4/GEOS-5 (http://www.acom.ucar.edu/wrf-chem/mozart.shtml). The aerosols are assumed internally mixed in this model. The number of aerosol bins is 5 covering from 0.01 to 10 $\mu$m. Further details about the model configuration may be found in Couvidat et al. (2012).

**Table 3.** List of the sensitivity simulations using different options in SOAP.

| Simulation name | Absorption approach | Activity coefficient | Viscous aerosol | Number of aerosol layers |
|---|---|---|---|---|
| Equilibrium UNIFAC | equilibrium | UNIFAC | No | 1 |
| Equilibrium Ideal | equilibrium | ideal mixture | No | 1 |
| Equilibrium UNIFAC-sr | equilibrium | UNIFAC-sr** | No | 1 |
| Equilibrium AIOMFAC | equilibrium | AIOMFAC | No | 1 |
| Dynamic inviscid | dynamic | UNIFAC | No | 1 |
| Dynamic viscous | dynamic | UNIFAC | Yes | 2 |

**: activity coefficients are calculated taking into account inorganic aerosols.





Six sensitivity simulations are conducted over Europe to study the effect of non-ideality and non-equilibrium phenomena on SOA formation. The list of the simulations is presented in Table 3. The reference simulation (named "Equilibrium UNIFAC") uses the default model options, which leads to the lowest computational time : thermodynamic equilibrium between the gas and particle phases is assumed and activity coefficients are computed with UNIFAC. To evaluate the impact of activity coefficients on concentrations, a simulation (named "Equilibrium Ideal") is run. The impact of inorganic aerosols on the short-range activity

coefficients using UNIFAC is estimated with a simulation (named "Equilibrium UNIFAC-sr"). To evaluate the impact of inorganic concentrations on activity coefficients, a simulation (named "Equilibrium AIOMFAC") using AIOMFAC to compute activity coefficients instead of UNIFAC is run.

To evaluate the impact of a viscous aqueous phase on SOA concentrations, two other simulations are run: one with a dynamic approach to model condensation/evaporation of inviscid particles (simulation named "Dynamic inviscid"), and one

with a dynamic approach but extremely viscous particles (simulation named "Dynamic viscous"). The simulations that use the equilibrium approach for condensation/evaporation are run from 1 June to 31 August 2012. However, the sensitivity simulations using the dynamic approach are run for only 3 weeks starting 1 June 2012, because of expensive computational time.

The used absorption approaches for the simulations are presented in Table 3. The absorbing mass includes inorganic aerosol, hydrophilic organic aerosol and water absorbed by inorganic aerosol/hydrophilic organic aerosol for $c_{aq}$. $c_p$ includes hydropho-

bic organic aerosol and water absorbed by hydrophobic organic aerosol as listed in Table 2.

### 4.2 Model evaluation

To evaluate the general performances of the model, the concentrations of organic aerosols given by the "Equilibrium UNIFAC" simulation are compared to the concentrations of organic matter (OM) and organic carbon (OC) measured at stations of the ACTRIS observation network (http://actris.nilu.no) in Europe. OC concentrations are measured by high/low volume samplers

and OM concentrations are measured by aerosol mass spectrometers. The locations of stations are presented in Figure 1. To compare the simulated OM concentrations with the measured OC, the simulated OM concentrations are converted into OC concentrations using the OM/OC ratio for each surrogate of the organic aerosols, as described in Couvidat et al. (2012).

To evaluate the model ability to reproduce SOA concentrations, the standard metrics of the model performance for particulate matter of Boylan and Russell (2006) are used: the mean fractional bias (MFB) and the mean fractional error (MFE). Boylan

and Russell (2006) proposed a performance evaluation criteria (|MFB| < 60% and MFE < 75%) and a goal evaluation criteria (|MFB| < 30% and MFE < 50%). Model performance statistics are presented in Table 4. For organic compounds, the model performance and goal criteria are satisfied for the stations Kosetice, Melpitz, Ispra, and Cabauw (|MFB| < 30% and MFE < 50%, see Figure 5). For the Birkenes station, the performance criteria are satisfied, but the goal criteria are almost satisfied, although the concentrations of $OM_1$ are quite underestimated (MFB: -33%). For the Aspvreten station, the model performance

criteria are satisfied but the $OC_{10}$ concentrations are significantly underestimated (MFB: -56%). For the Montseny stations, the goal criteria are satisfied for OC and the performance criteria are satisfied for $OM_1$ with a significant underestimation.



**Table 4.** Comparison of the simulated concentrations to the measurements. Performance statistics are calculated with daily mean concentrations.

| Station | Particle type | Measurement[†] ($\mu g\,m^{-3}$) | Simulation[†] ($\mu g\,m^{-3}$) | RMSE ($\mu g\,m^{-3}$) | MFB | MFE | Correlation |
|---|---|---|---|---|---|---|---|
| Kosetice | $OC_{2.5}$ | 2.36 | 1.94 | 0.86 | -26% | 37% | 0.73 |
| Melpitz | $OC_{2.5}$ | 1.41 | 1.67 | 0.56 | 16% | 27% | 0.85 |
| | $OC_{10}$ | 2.05 | 1.67 | 0.63 | -23% | 28% | 0.86 |
| | $OM_1$ | 3.83 | 2.75 | 2.21 | -20% | 38% | 0.81 |
| | $NH_{4,1}$ | 0.66 | 0.46 | 0.32 | -30% | 36% | 0.72 |
| | $NO_{3,1}$ | 0.84 | 0.53 | 0.59 | -47% | 60% | 0.56 |
| | $SO_{4,1}$ | 1.60 | 0.84 | 0.91 | -62% | 63% | 0.74 |
| Montseny | $OC_1$ | 1.89 | 2.06 | 0.57 | 6% | 20% | 0.76 |
| | $OC_{2.5}$ | 2.41 | 2.20 | 0.70 | -10% | 23% | 0.66 |
| | $OC_{10}$ | 2.72 | 2.06 | 0.99 | -30% | 35% | 0.64 |
| | $OM_1$ | 7.56 | 3.90 | 4.80 | -60% | 64% | 0.26 |
| | $NH_{4,1}$ | 1.14 | 0.59 | 0.78 | -60% | 60% | 0.49 |
| | $NO_{3,1}$ | 0.58 | 0.36 | 0.77 | -51% | 66% | 0.04 |
| | $SO_{4,1}$ | 2.54 | 1.23 | 1.84 | -65% | 66% | 0.44 |
| Ispra | $OC_{2.5}$ | 2.79 | 3.16 | 0.88 | 16% | 27% | 0.75 |
| Aspvreten | $OC_{10}$ | 2.26 | 1.29 | 1.14 | -56% | 56% | 0.63 |
| Birkenes | $OM_1$ | 1.55 | 1.07 | 0.54 | -39% | 39% | 0.80 |
| Cabauw | $OM_1$ | 2.86 | 2.59 | 1.11 | 0% | 20% | 0.94 |
| | $NH_{4,1}$ | 1.04 | 1.09 | 0.52 | 11% | 30% | 0.78 |
| | $NO_{3,1}$ | 2.82 | 2.65 | 2.00 | 12% | 49% | 0.75 |
| | $SO_{4,1}$ | 0.88 | 0.95 | 0.31 | 8% | 21% | 0.79 |

[†]: mean concentration from June 1 to August 31, 2012.

For inorganic $PM_1$ aerosols, the performance biases and errors are satisfied at the Cabauw station. However, inorganic $PM_1$ concentrations are underestimated at the Montseny and Melpitz stations, even though the MFE satisfies the model performance criteria.

## 4.3 Impact of inorganic-organic interactions

Figure 6 shows the temporal evolution of the domain-averaged concentration of hydrophilic SOA for different simulations. Modeling organic interactions by activity coefficients strongly influences hydrophilic SOA. It leads on average to a concentration increase of 33% (simulation "Equilibrium-UNIFAC" compared to simulation "Equilibrium-Ideal"). When the computation





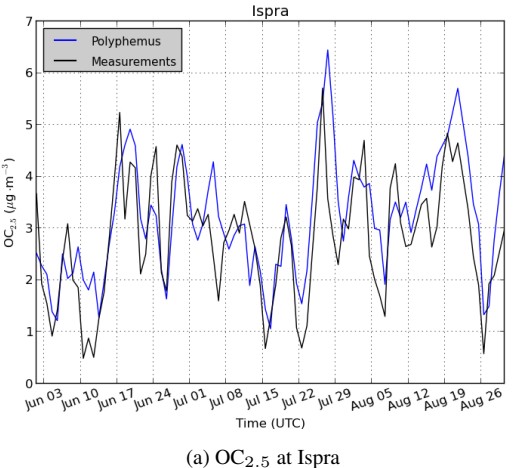
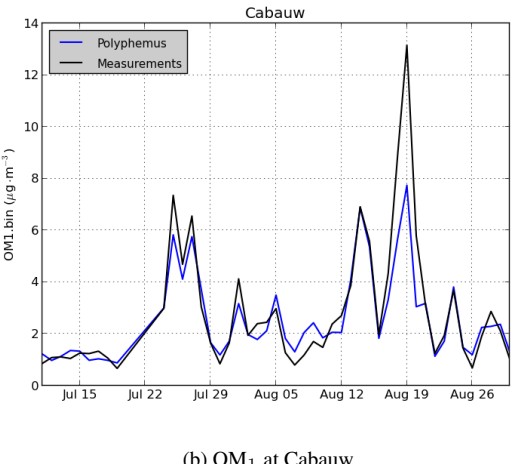

(a) $OC_{2.5}$ at Ispra        (b) $OM_1$ at Cabauw

**Figure 5.** Comparison of modeled SOA concentrations (blue) with observations (black) for (a) $OC_{2.5}$ concentrations at Ispra, (b) $OM_1$ concentrations at Cabauw.

of short-range interactions between inorganic and organic aerosols is taken into account in UNIFAC, the SOA concentrations increase because of a decrease of activity coefficient (see "Equilibrium UNIFAC-sr" simulation in Figure 6d). Long-range and medium-range interactions in the "Equilibrium AIOMFAC" simulation lead to an increase of activity coefficients as concentrations decrease compared to the "Equilibrium UNIFAC-sr" simulation by 28%. This evaporation of hydrophilic organic concentrations by the medium and long-range inorganic-organic interactions agrees with the results of Zuend et al. (2008), who showed that the activity coefficients of hydrophilic organic aerosols increase because of the interactions with inorganic aerosols.

5    The SOA concentrations simulated with the AIOMFAC model are close to the concentrations simulated with the UNIFAC model (without taking into account inorganics in the computation of short-range interactions). It suggests that computing activity coefficients for hydrophilic organic compounds by only taking water and organic compounds (and therefore by ignoring inorganics) could give a good first approximation of activity coefficients. Medium and long-range interactions compensate the decrease of activity coefficients due to the inclusion of inorganic ions in short-range interactions. We estimated the contributions

10   of long-range and medium-range interactions in this decrease by an additional simulation. In this additional simulation, only the medium-range interactions are taken into account in the AIOMFAC model. According to the results of this simulation, the differences in the concentrations of hydrophilic SOA are due to the medium-range interactions by 65% and the long-range interactions by 35%.

    However, the choice of thermodynamic model affects the spatial distribution of hydrophilic SOA with an decrease of concen-

15   trations when using AIOMFAC over Netherlands, Belgium, part of Italy, Spain over the Mediterranean coast and southeastern Europe and an increase of concentrations over Northern Europe, part of the Alps, southern France, part of Italy and part of Spain. The area with the strongest decrease of concentrations corresponds to areas with strong inorganic concentrations. For





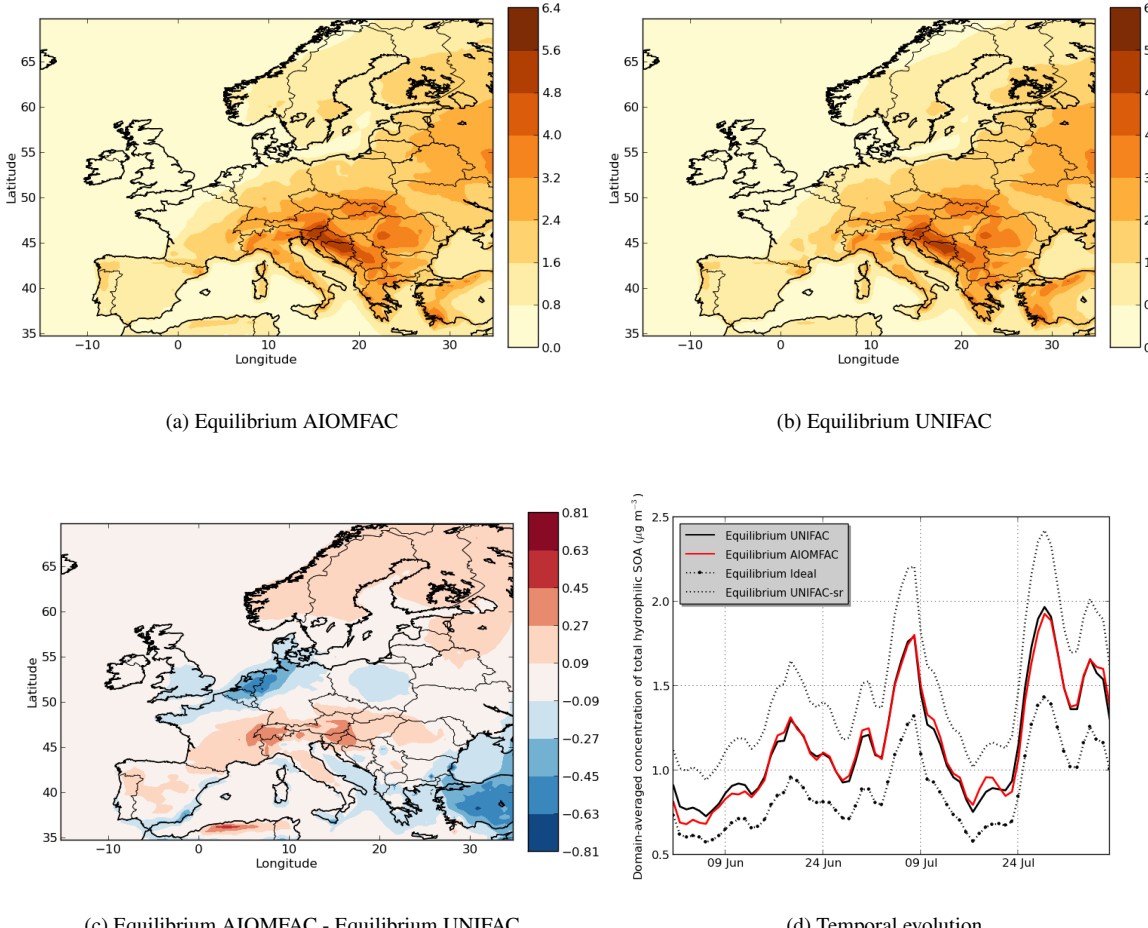

(a) Equilibrium AIOMFAC

(b) Equilibrium UNIFAC

(c) Equilibrium AIOMFAC - Equilibrium UNIFAC

(d) Temporal evolution

**Figure 6.** Modeled hydrophilic SOA concentrations in (a) the Equilibrium AIOMFAC simulation, (b) the Equilibrium UNIFAC simulation, (c) the differences between the simulations ($\mu g\,m^{-3}$) and (d) the temporal evolutions of domain averaged concentrations.

example, the decrease of concentrations over Netherlands corresponds to high ammonium nitrate concentrations while the decrease in southeastern Europe corresponds to high ammonium sulfate concentrations.

The changes in concentration of specific SOA compounds using AIOMFAC and UNIFAC are illustrated by Figure 7. The local increases of concentrations can be due to non-linear effects. Indeed, while the concentrations of the less oxidized hydrophilic compounds (BiA0D with a O/C of 0.2 and BiA1D with a O/C of 0.5) mainly decrease over Europe (-2% for BiA0D and -27% for BiA1D), the concentrations of the more oxidized compounds (BiPER with a O/C of 1.2 and BiDER with a O/C of 0.8) mainly increase (6% for BiPER and 16% for BiDER). This finding is in agreement with Pye et al. (2018) who found





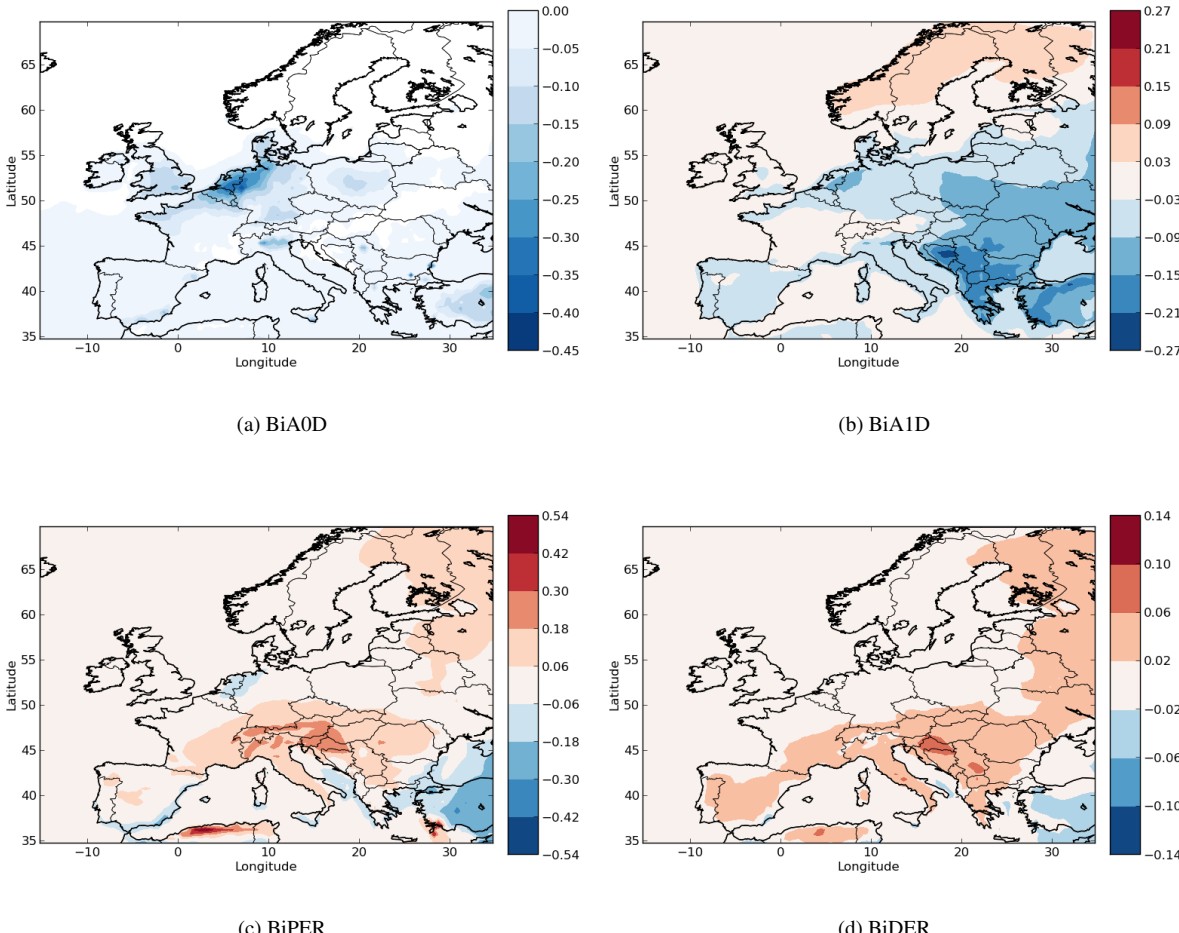

**Figure 7.** Differences in concentrations of several SOA compounds between AIOMFAC and UNIFAC over Europe (AIOMFAC - UNIFAC in µg m$^{-3}$). The definition of the compounds is given in Table 1.

that in the eastern US, particle-phase interactions of organic and inorganic compounds increase partitioning toward the particle phase (vs. gas phase) for highly oxidized compounds (O:C $\geq$ 0.6) but decrease particle-phase partitioning for low O:C.

## 4.4 Impact of the thermodynamic equilibrium assumption

The aqueous phase of the particles is assumed inviscid and organics are strongly influence by inorganic concentrations, because they constitute an absorbing mass for hydrophilic organics. However, in the organic phase, the particles may be viscous, and the dynamic evolution of the SOA concentrations by condensation/evaporation may be limited by diffusion due to the particle viscosity (Couvidat and Sartelet, 2015).





5   To evaluate the impact of particle viscosity on SOA concentrations, condensation/evaporation need to be solved using the dynamic approach. Because condensation/evaporation is solved using the equilibrium approach in the previous simulations, the impact of using the dynamic approach, while still assuming particles to be inviscid, is assessed by running the "Dynamic inviscid" simulation.

Differences between the "Equilibrium-UNIFAC" and the "Dynamic inviscid" simulations are very low for hydrophobic compounds (less than 1%) whereas a decrease of concentrations by about 6% is found for hydrophilic compounds in the "Dynamic inviscid" simulation. The differences are due to the non-ideality of the aerosols as low differences are found when assuming ideality (3%). In the "Equilibrium-UNIFAC" simulation, activity coefficients are computed by taking the composition

5   of the aerosols averaged over size sections. However, for the "Dynamic inviscid" simulation, activity coefficients are computed for each size section. The section activity coefficients of the "Dynamic inviscid" simulation are therefore different from the activity coefficients of the "Equilibrium-UNIFAC" simulation.

For hydrophobic compounds, the differences are mainly due to the variations of the mass transfer rate computed by Equation 5. In the dynamic approach, the condensation/evaporation is slower than in the equilibrium approach. Therefore, using the dynamic approach reduces the magnitudes of the peaks in the temporal variations of the SOA concentrations, although the average concentrations do not change much with the temporal and spatial resolutions used here. In the dynamic approach, in

opposition to the equilibrium approach, low-volatility secondary compounds formed by gas-phase chemistry are found to not be totally into the particle phase due to the kinetic of condensation. For example, 97% of SOAlP is absorbed inside the particle and 3% of SOAlP is still present in the gas-phase.

### 4.5   Impact of viscosity of the organic phase

In the simulation Dynamic-viscous, as expected, the dynamic evolution of hydrophilic SOA concentration does not change

from those of the "Dynamic inviscid" simulation, but the organic hydrophobic phase is strongly influenced by the viscosity.

Assuming that the organic phase is very viscous leads to an increase in concentrations of hydrophobic SOA: 6% on average of the total concentrations (see Figure 8). The increase can exceed 20% over areas with low concentrations in the "Dynamic inviscid" simulation (Spain and Northern Europe). This increase of concentrations of hydrophobic SOA is due to the absence of evaporation (because of the absence of diffusion) when concentrations exceed equilibrium. The hydrophobic SOA concen-

trations in the "Dynamic viscous" simulation decrease where they are very high in the "Dynamic inviscid" simulation (the Straits of Gibraltar and Istanbul). As shown in Figure 8d, the increase of concentrations happen mainly during daytime.

The influence of viscosity differs depending on the volatility of the surrogate. For example, in the model, the emitted anthropogenic I/S-VOC are represented by surrogates of different volatility classes: high volatility (POAhP, $K_p = 0.031$ m$^3$ $\mu$g$^{-1}$), average volatility (POAmP, $K_p = 0.0116$ m$^3$ $\mu$g$^{-1}$) and low volatility (POAlP, $K_p = 1.1$ m$^3$ $\mu$g$^{-1}$). The chemical kinetic mech-

anism used for the SOAP model includes the oxidation of these surrogates to other surrogates with lower volatilities: SOAhP ($K_p = 0.031$ m$^3$ $\mu$g$^{-1}$), SOAmP ($K_p = 1.16$ m$^3$ $\mu$g$^{-1}$), SOAlP ($K_p = 110$ m$^3$ $\mu$g$^{-1}$) with the following equations (Couvidat et al., 2012).

POAhP + OH $\longrightarrow$ SOAhP

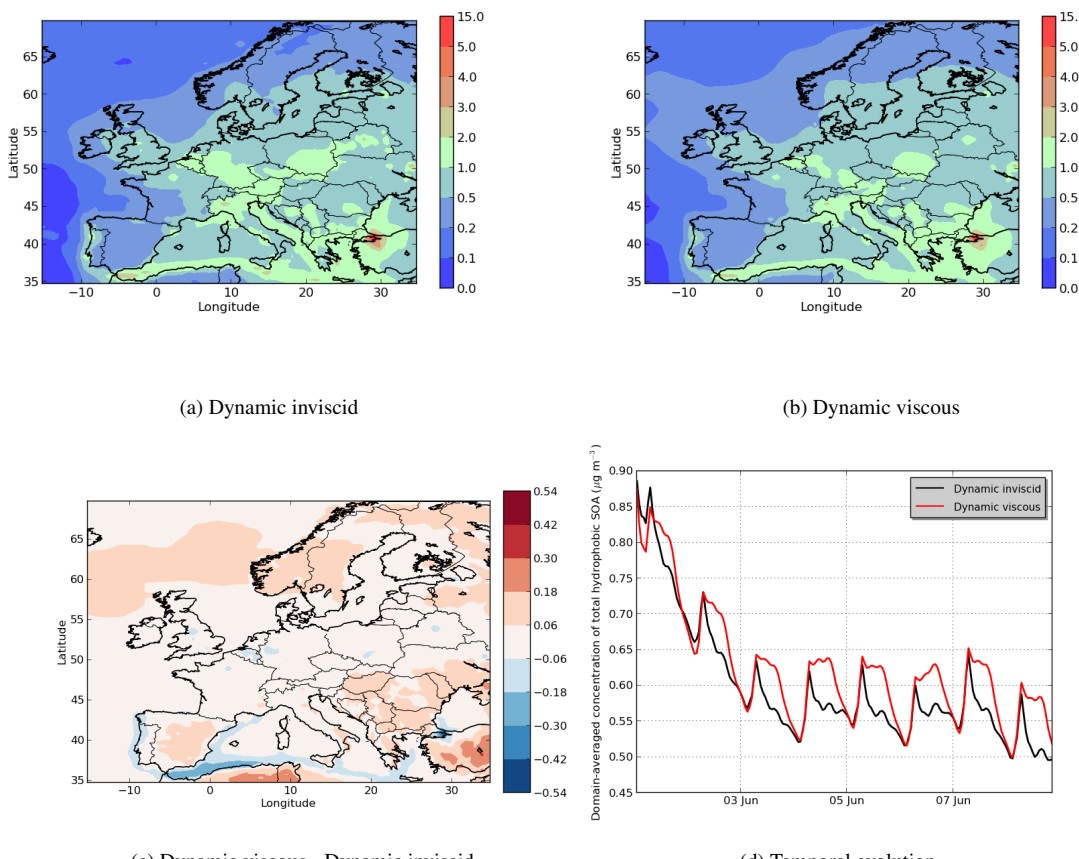

(a) Dynamic inviscid

(b) Dynamic viscous

(c) Dynamic viscous - Dynamic inviscid

(d) Temporal evolution

**Figure 8.** Modeled hydrophobic SOA concentrations in (a) the Dynamic inviscid simulation, (b) the Dynamic viscous simulation, (c) the differences between the simulations ($\mu g\ m^{-3}$) and (d) the temporal evolutions.

$$POAmP + OH \longrightarrow SOAmP$$

$$POAlP + OH \longrightarrow SOAlP$$

In Figure 9a, the concentrations of SOAhP (one of the most volatile compounds of the mechanism) strongly increase in the "Dynamic-viscous" simulation (by 44% in average). This increase is especially strong in Southeastern Europe where concentrations double and increase by 0.1 $\mu g\ m^{-3}$. In the "Dynamic viscous" simulation, concentrations increase strongly at the beginning of the day and reach a maximum during daytime. On the contrary, in the "Dynamic inviscid" simulation,

concentrations decrease at the beginning of the day and reach a minimum during daytime (as the volatility of the compound increases during daytime). In the "Dynamic viscous" simulation, the diurnal variations of SOAhP follow those of the low-volatility compound SOAlP (Figure 9b). Figure 10 show the deviation of the particle/gas partitioning from equilibrium for



(a) SOAhP

(b) SOAlP

(c) POAlP

**Figure 9.** Temporal evolution (left) and differences between the Dynamic viscous and Dynamic inviscid simulations (right) of modeled SOA concentrations for (a) SOAhP, (b) SOAlP, and (c) POAlP ($\mu$g m$^{-3}$).





SOAhP (defined as the particle/gas concentration ratio over the particle/gas ratio for the inviscid simulation, which is close to equilibrium). This deviation often exceeds 50% and particle-phase concentrations exceeds equilibrium over most of Europe. As presented in section 3.3, condensation of a semi-volatile compound can happen without respecting equilibrium as long as the particle is growing (growth that can be due to the condensation of a less-volatile compound such as SOAlP). The condensation during the day of non-volatile compounds formed during daytime stops the evaporation of SOAhP captured inside the particle (evaporation that would occur for an inviscid organic phase) and is even able to bring further condensation of the compound.

Concentrations for the non-volatile compound SOAlP slightly decrease (see Figure 9b). This decrease is mainly due to the increase of the POAlP particle-phase concentration during daytime (see Figure 9c). In the chemical kinetic mechanism used in this study, SOAlP is formed from the gas-phase oxidation of POAlP by OH radical (mainly present daytime). The increase of POAlP in the particle-phase during daytime slows down the formation of the compound SOAlP. On the contrary, at the end of the day, concentrations of POAhP become higher in the "Dynamic inviscid" simulation due to the decrease of volatility (because of the decrease of temperature). However, in the "Dynamic viscous" simulation, the decrease of the volatility has a low effect on concentrations (because the internal layer cannot absorb more compounds to reach equilibrium due to the absence of diffusion).

The large deviations from equilibrium suggested by this study agree with the measurements of Yatavelli et al. (2014) and Lopez-Hilfiker et al. (2015) who observed that the concentrations of pinonic acid in SOA are much higher than predicted with the equilibrium assumption using saturation vapor pressures. It could also be possible that this phenomenon is due to non-ideal effect and the possibility for pinonic acid to be absorbed onto an aqueous phase with an acidic dissociation.

## 4.6 Comparison of computation times

Table 5 presents the time elapsed for each simulation. The elapsed time for the "Equilibrium UNIFAC" simulation is set to a reference time. The time elapsed for the "Equilibrium AIOMFAC" simulation increases by 45% compared to the reference time. Using the dynamic approach leads to an increase of the computation time by a factor ten, making it difficult to represent viscous aerosols in long-term 3-D simulations. However, these computation times are acceptable for short-term 3-D simulations.

**Table 5.** Comparison of time elapsed for the simulations. The elapsed time for the Equilibrium UNIFAC simulation is set to a reference time and ratios between the reference time and the time elapsed for other simulations are presented.

|  | Equilibrium Ideal | Equilibrium UNIFAC | Equilibrium AIOMFAC | Dynamic Inviscid | Dynamic Viscous |
|---|---|---|---|---|---|
| Ratio of computation time | 0.97 | 1 | 1.45 | 9.74 | 10.02 |





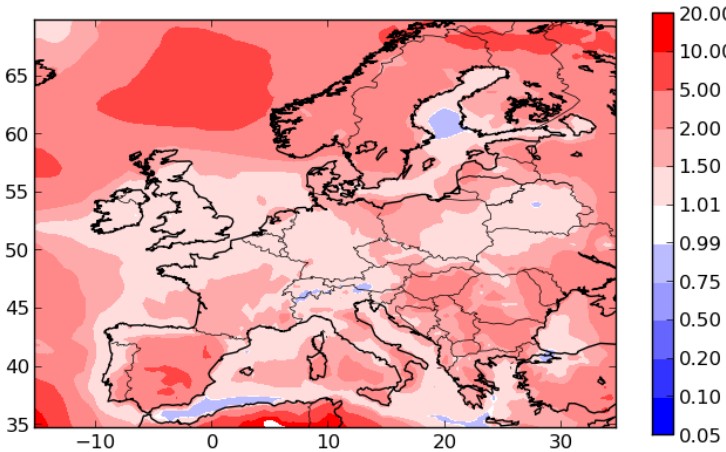

**Figure 10.** Equilibrium deviation for compound SOAhP. A deviation close to 1 means that the compound reaches equilibrium, above 1 means that particle-phase concentrations are above equilibrium and under 1 that concentrations have not reached equilibrium.

## 5   Conclusions

The SOAP model, which uses either the equilibrium approach or the dynamic approach for the mass transfer of organic compounds from the gas phase to the particle phases, was implemented in the 3-D air quality model of Polyphemus. Compared to its predecessor, SOAP provides a more complete description of the partitioning of semi-volatile compounds, in particular, by taking into account the effect of inorganic aerosols on SOA formation based on the computation of activity coefficients given by AIOMFAC. Sensitivity simulations indicate that including inorganic aerosols and hydrophilic organic aerosols in the absorbing mass of the aqueous-phase can lead to an increase of concentrations around 5% and 6%, respectively. Overall, hydrophilic SOA concentrations using AIOMFAC are higher than those with the ideality assumption by about 33%. The results of this study suggest that non-ideality via organic-organic and inorganic-organic interactions influence strongly the condensation of hydrophilic organic compounds.

For an inviscid aerosol, the results of this study show that the equilibrium assumption is an efficient approximation, when assuming ideality for organic aerosols. However, assuming equilibrium can lead to significant differences in the concentrations of hydrophilic compounds when non-ideality is taken into account. Indeed, with a dynamic approach, different values of activity coefficients can be simulated for the different size sections. These results indicate that differences in the composition



of particles with particle size can impact the formation of SOA. Note that in this study, an equilibrium approach is used for the condensation of inorganics. Using a dynamic approach to model the condensation/evaporation of both inorganic and organic compounds may be necessary to properly estimate the formation of hydrophilic SOA.

The dynamic approach in the SOAP model is used to account for the viscosity of aerosol to study SOA formation via two theoretical cases: the "inviscid case" where diffusion is extremely fast and concentrations inside the particle are homogeneous and the "infinite viscosity case" where viscosity is too high for diffusion to occur inside the particle but where condensation or evaporation of compounds at the gas/particle interface can still occur. Even if the two cases presented in this study are theoretical, the results provide a first insight on how viscosity may affect SOA formation. For the "inviscid case", concentrations of hydrophobic compounds are close to those in the equilibrium simulation. However, assuming a highly viscous organic-phase leads to an increase of hydrophobic SOA concentration during daytime (by stopping the evaporation of the most volatile compounds without stopping their condensation). SOA formation for a highly viscous particle can therefore significantly deviate from thermodynamic equilibrium, e.g., condensation can happen when evaporation occurs if equilibrium is assumed. This deviation may explain why some observed concentrations in the literature are significantly different to concentrations calculated with the equilibrium assumption and saturation vapor pressures.

Those results emphasize the need to study the effect of the dynamics of SOA formation. Next modeling studies should focus on the sensitivity of results to the organic-phase diffusion coefficient and try to take into account the effect of temperature, the aerosol water content and also aerosol composition on this parameter.

The estimation of the computation time shows that the dynamic approach used in the SOAP model can be applicable to the 3-D air quality modeling for a short period or with high computation time capability. Although, the results emphasize the need to study the effect of a dynamic approach compared to an equilibrium approach, the computation-time issue is probably a limiting factor in the possibility for the implementation of dynamic approaches in 3-D air quality models.

Finally, the effect of morphology for a highly viscous aerosol may be critical for a highly viscous aerosol. The coagulation of two highly viscous spherical particles may form a non-spherical particle composed of two spheres stuck together. Non-spherical particles may lead to higher surface on volume ratio and faster condensation/evaporation/diffusion.

*Acknowledgements.* This work was funded by the French Minister of Environment and Sustainable Development under the project NA-TORGA of the COPERNICUS-MDD program. The authors would like to thank the principal investigators of the ACTRIS measurements used for the model evaluation (Rupert Holzinger, Patrick Schlag, Astrid Kiendler-Scharr for the station Cabauw; Wenche Aas, Chris Lunder for the station Birkenes; MariaCruz Minguillon, Anna Ripoll, Andres Alastuey for the station Montseny; Laurent Poulain for the station Melpitz; F. Cavalli and J.P. Putaud for the station Ispra; Milan Vana for the station Kosetice; Peter Tunved, Hans Areskoug for the station Aspvreten).



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
