# Peer review of "Modeling the effect of non-ideality, dynamic mass transfer and viscosity on SOA formation in a 3-D air quality model"

_Atmospheric Chemistry and Physics, 2018_

## Referee Comment (RC1) · Anonymous Referee #2 · 12 Jul 2018

This manuscript addresses the presentation of secondary organic aerosols in air quality model, specifically the non-ideality and viscosity of the particles. This is an important topic on the field of SOA studies and the topic is well suited for ACP. I find some points that should be addressed before publishing in ACP.

Specific comments:

Authors present organics with few surrogate compounds and this is understandable approach in 3D air quality model. There is no discussion in the manuscript about the effect of choice of the surrogate compounds on the results and I wonder if the authors could comment this little bit. First, how does the use of 21 surrogate compounds

(instead of the huge number of organics in the real atmosphere) affect the results regarding the non-ideality and the activity coefficient calculations; Is it expected that activity coefficients matter that much also when the particles consist of mixtures of much more compounds? Second, how does the choice of surrogate compounds affect the viscosity effect? For instance, monoterpene oxidation products seem to be missing low-volatility compounds (unless there is acidic aqueous phase). If the mass of low-volatility compounds was underestimated, wouldn't that lead to viscosity effect being overestimated?

There are some aspects in the description of dynamic calculations for viscous particles where some clarification would be helpful: 1) Eq. 7 seems to describe the diffusion in particle phase based on difference between equilibrium particle phase concentration and the actual particle phase concentration of compound i. Based on the given reference (Couvidat and Sartelet 2015), in eq. 7 the concentrations should be for each layer, not the total particle phase concentrations, right? 2) The equation in the given reference seems to be derived for a case where the particle phase concentration at the interface is constant (equilibrium with gas phase). This is not the case here if also condensation/evaporation is calculated dynamically. I wonder if the equation is correct to use here. 3) If the J_diff in eq. 5 is calculated for each layer as in the reference, then it is not clear which J_diff is used in eq. 10. for calculating J_tot. 4) Only two layers are used for calculations here. Is the model accurate with only two layers? In the given reference the model is evaluated only for three or more layers.

It is stated (P14, L20) that the aerosols are assumed internally mixed, but still different absorbing masses are used for hydrophilic and hydrophobic compounds. Are the two phases assumed to co-exist in a particle? How is this taken in to account in the dynamic approach where equations for spherically symmetrical cases are used (eq. 5-9)?

Figure 4 "Shrinking" part and related discussion on P14 L6-9: It is not clear how shrinking in this conceptual discussion is thought to happen. Could the authors clarify this? Especially confusing is the blue text about volatile compound A being stuck at the interface due to shrinking. In case of a net mass flux of A towards the particle, A could be stuck at the interface and not transferred to the particle core. But how does that happen at the same time when A is evaporating and the particle thereby shrinking?

Technical comments:

Table 1, row BiA0D, last column: A typo: "qaueous".

Table 1: Units missing for Kp.

P8, L15: Text refers to "SOAP-basic" although such simulations is not presented in the fig. 2.

P10, L18: "concentrations" twice.

---

## Referee Comment (RC2) · Anonymous Referee #1 · 31 Jul 2018

Kim et al. investigated the effects of non-ideality, dynamic mass transfer and aerosol viscosity on SOA formation using an air quality model. Most of the current 3-D air quality models have not explicitly considered the role of kinetic partitioning and the interactions between organic and inorganic aerosol components. The organic aerosol community could certainly benefit from this modeling study. However, it is recommended that the manuscript be significantly improved before being published in ACP.

Major comments: (1) My major concern is related to the assumptions of particle viscosity. In the "Dynamic viscous" scenario, the particle diffusivity is assigned as a constant value, as low as 10-30 m2 s-1. I wonder is it necessary to assign such a small value

that almost stopes the bulk diffusion, which would rarely happen in the ambient air. In addition, particle viscosity should be a function of temperature, RH, and particle composition. However, in the sensitivity simulations in Section 4.5, the particle diffusivity is kept constant. Thus, it is hard to convince the readers that the results presented in Section 4.5 could reflect the impact of particle viscosity on SOA formation / evaporation in real atmospheric conditions.

(2) Although the development of the SOAP module has been published in Couovidat and Sartelet (2015), some treatments should be clarified when coupling SOAP into a 3-D model. Firstly, in Couovidat and Sartelet (2015), ISORROPIA is called prior to SOAP and the amount of particle water will adopt the value computed by ISORROPIA instead of SOAP when the water in SOAP is lower than the water in ISORROPIA. This treatment will cause uncertainties in particle water calculations as stated in Couovidat and Sartelet (2015). Has ISORROPIA been fully coupled with SOAP in the current 3-D modeling? If not, what is its impact on the results of SOA partitioning? I suggest adding a spatial distribution of particle water in Figure 3. Secondly, in Couovidat and Sartelet (2015), when treating the diffusion of organic compounds in spherical organic particles, it is assumed that the concentrations in one layer can be described independently from the concentrations in the other layers and the mass fraction (ratio of the mass of the layer over the mass of the particle) of layers must stay constant throughout the simulation. It seems the authors followed Couovidat and Sartelet (2015) and assumed that the transfer between the interface and the internal layer "can happen without diffusion to assure that the mass fraction of layers remain constant even during the growth or shrinking of the particle" (Page 13). This assumption may explain the unreasonable blue arrows in Figure 4 which is also questioned by the other referee. I agree with the related comments proposed by the other referee on Figure 4 and suggest revising this treatment in SOAP as the results in Figure 4 are contrary to previous kinetic simulations (please refer to Liu et al. PNAS, 2016; Mai et al. EST, 2015; Shiraiwa et al. GRL, 2012). For example, in Fig.4a, why compound A in the particle phase can not transfer from the core to the interface while compound A in the gas phase can transfer from

the interface to the core? Is it just because the mass ratios of the layers have to be maintained constant as the authors stated in the last paragraph in Section 3.3?

Thirdly, what is the time step adopted in the 3-D model? This is related whether the gas-particle partitioning has been reached equilibrium within the time step set in the 3-D model. Fourthly, the accommodation coefficient $\alpha$ is assumed as 0.5 in this study. Will the simulation results be changed if $\alpha$ is adopted as 1 (see the recent experiment study by Krechmer et al. 2017). Lastly, I agree with the comments proposed by the other referee that the authors stated that "the aerosols are assumed internally mixed" (Section 4.1) but the absorbing phases are still calculated separately (caq and cp stated in the last paragraph in Section 4.1).

(3) I have some concern about the presentation quality. For example: the title of Section 2.1 is "Composition of aerosols" but Section 2.1 is about gas-particle partitioning. In Section 2.1, the authors firstly stated that in SOAP the cp included cwater,p, while in the following Eq(2) cwater,p is calculated from cp. I am confused in Eq(2) cp includes cwater,p or not. Figure 2: the scenario of "SOAP-basic" is missing although it is compared with "SOAP-no_water" in the text (Page 8). Figure 2 shows the diurnal variations, but the text lacks the description why the differences of simulated SOA concentrations are sometimes large in daytime while sometimes are large in nighttime. Page 10 Section 3.1, the authors stated that the inorganic compounds influence activity coefficients "by middle-range and long-range interactions" but as already stated on Page 8, inorganic compounds also influence the short-range activity coefficient. Last sentence on Page 11, change "viscous compounds" to "viscous particles". Last sentence on Page 13, "the compound A condenses onto the new layer created by the compound B" is not true. In my understanding, the layers in the model are kept same during SOA formation / evaporation. Section 4.2, besedes MFE and MFB, RMSE should also be defined. OM1 etc in Table 4 should also be defined. Section 4.3, it is better to describe Fig 6 from Fig 6a to Fig 6d instead of beginning from Fig 6d. The first sentence in Section 4.4, "influence" should be "influenced".

References: - Krechmer et al. 2017, Direct Measurements of Gas/Particle Partitioning and Mass Accommodation Coefficients in Environmental Chambers, Environmental Science & Technology, 51(20), 11867-11875.

- Liu, P., Y. J. Li, Y. Wang, M. K. Gilles, R. A. Zaveri, A. K. Bertram, and S. T. Martin (2016), Lability of secondary organic particulate matter, Proceedings of the National Academy of Sciences, 113(45), 12643-12648.

- Mai, H., M. Shiraiwa, R. C. Flagan, and J. H. Seinfeld (2015), Under What Conditions Can Equilibrium Gas–Particle Partitioning Be Expected to Hold in the Atmosphere?, Environmental Science & Technology, 49(19), 11485-11491.

- Shiraiwa, M., and J. H. Seinfeld (2012), Equilibration timescale of atmospheric secondary organic aerosol partitioning, Geophysical Research Letters, 39(24).

---

## Author Comment (AC1) · 9 Oct 2018

We appreciate the reviewers for reading the manuscript attentively and giving helpful comments to improve our manuscript.

**1   Reply to anonymous referee #1's comments**

**General comments**

Kim et al. investigated the effects of non-ideality, dynamic mass transfer and aerosol

viscosity on SOA formation using an air quality model. Most of the current 3-D air quality models have not explicitly considered the role of kinetic partitioning and the interactions between organic and inorganic aerosol components. The organic aerosol community could certainly benefit from this modeling study. However, it is recommended that the manuscript be significantly improved before being published in ACP.

**Major comments**

1. My major concern is related to the assumptions of particle viscosity. In the "Dynamic viscous" scenario, the particle diffusivity is assigned as a constant value, as low as $10^{-30}$ m$^2$ s$^{-1}$. I wonder is it necessary to assign such a small value that almost stopes the bulk diffusion, which would rarely happen in the ambient air. In addition, particle viscosity should be a function of temperature, RH, and particle composition. However, in the sensitivity simulations in Section 4.5, the particle diffusivity is kept constant. Thus, it is hard to convince the readers that the results presented in Section 4.5 could reflect the impact of particle viscosity on SOA formation / evaporation in real atmospheric conditions.

    **Our response**:

    It is difficult to estimate the particle viscosity. We understand that it depends on temperature, relative humidity and particle composition. Very recently, papers that could lead to some parameterization of viscosity came out to model viscosity (e.g., DeRieux et al., 2018). However, this is not straightforward to model. We would need to increase the number of particle layers, and the CPU performances would be very high. As a first approach, we decided to estimate the maximum potential contribution of viscosity by comparing extremely high and low values. The diffusivity of organic species is modeled using a bulk viscosity of the mixture estimated by the Refutas method (Maples, 2000). For the "Dynamic viscous" simulation, a very high viscous condition is used to investigate the maximum deviation of SOA concentrations from the inviscid condition. According to

measurement studies, the diffusivity of organic species in SOA can be lower than $10^{-21}$ m$^2$ s$^{-1}$ (e.g., Pfrang et al., 2011; Abramson et al., 2013). Song et al. (2016) and DeRieux et al. (2018) showed that scaled values from measured viscosities can pass through a viscosity of $10^{12}$ Pa s, which is the order of a diffusivity of $10^{-30}$ m$^2$ s$^{-1}$, at low relative humidity.

In addition, Couvidat and Sartelet (2015) reported that at a diffusivity of $10^{-24}$ m$^2$ s$^{-1}$, diffusivity does not influence the mass of the condensed organic species as the diffusion is too low to significantly affect the formation of organic aerosol that still occur by condensation/evaporation of organic compounds at the interface. Therefore a diffusitivy lower than $10^{-24}$ m$^2$ s$^{-1}$ may not affect the concentrations of organic aerosols compared to simulation results with a diffusivity of $10^{-24}$ m$^2$ s$^{-1}$.

The text has been modified in the revised manuscript as follows:

" A very low diffusivity of $10^{-30}$ m$^2$ s$^{-1}$ is assumed in order to investigate the maximum deviation of SOA concentrations from the inviscid condition. The diffusivity of organic species is modeled using a bulk viscosity of the mixture estimated by the Refutas method (Maples, 2000). According to measurement studies, the diffusivity of organic species in SOA can be lower than $10^{-21}$ m$^2$ s$^{-1}$ (e.g., Pfrang et al., 2011; Abramson et al., 2013). Song et al. (2016) and DeRieux et al. (2018) showed that scaled values from measured viscosities and predicted values can pass through a viscosity of $10^{12}$ Pa s, which is the order of a diffusivity of $10^{-30}$ m$^2$ s$^{-1}$, at low relative humidity. In addition, Couvidat and Sartelet (2015) reported that at a diffusivity of $10^{-24}$ m$^2$ s$^{-1}$, diffusivity does not influence the mass of the condensed organic species as the diffusion is too low to significantly affect the formation of organic aerosol that still occur by condensation/evaporation of organic compounds at the interface. Therefore a diffusitivy lower than $10^{-24}$ m$^2$ s$^{-1}$ may not affect the concentrations of organic aerosols compared to simulation results with a diffusivity of $10^{-24}$ m$^2$ s$^{-1}$. "

2. Although the development of the SOAP module has been published in Couvidat and Sartelet (2015), some treatments should be clarified when coupling SOAP into a 3-D model. Firstly, in Couvidat and Sartelet (2015), ISORROPIA is called prior to SOAP and the amount of particle water will adopt the value computed by ISORROPIA instead of SOAP when the water in SOAP is lower than the water in ISORROPIA. This treatment will cause uncertainties in particle water calculations as stated in Couvidat and Sartelet (2015). Has ISORROPIA been fully coupled with SOAP in the current 3-D modeling? If not, what is its impact on the results of SOA partitioning? I suggest adding a spatial distribution of particle water in Figure 3.

**Our response**:

The coupling of inorganic and organic aerosol formation may consist of two parts: one takes into account the organic species for the inorganic aerosol formation and the other one takes into account the inorganic species for the organic aerosol formation. The latter has been implemented in the SOAP model and in the 3-D model. However, for the former, a new inorganic model including the organic species need to be developped and coupled with the SOAP model. This is out of the scope of this work.

For the potential impacts, as mentionned in Couvidat and Sartelet (2015), the organic species can either reduce or enhance the water absorption of inorganic species (Choi and Chan, 2002). This changed water content can lead to a change in the condensation of organic species.

Following the reviewer's suggestion, Figure 1 (Figure 3c in the revised manuscript) and text for averaged concentration of water have been added to the revised manuscript.

" The coupling of inorganic and organic aerosol formation influences the water absorption by particles. This coupling consists of two effects: the influence of organic species on the inorganic aerosol formation and the influence of inorganic

species on the organic aerosol formation. The latter is implemented in the SOAP model, but the influence of organic species on the inorganic aerosol formation is not included. According to Choi and Chan (2002), the organic species can either reduce or enhance the water absorption of inorganic species, which in turns can lead to a change in the condensation of organic species.

Figure 3c shows the concentration of total condensed water in the SOAP-Reference simulation. The coupling of inorganic and organic aerosol formation may lead to changes in the aerosol concentrations in the regions where both the concentrations of total condensed water and hydrophilic organic species are large, e.g., Barcelona, Milano and Eastern Spain. "

3. Secondly, in Couvidat and Sartelet (2015), when treating the diffusion of organic compounds in spherical organic particles, it is assumed that the concentrations in one layer can be described independently from the concentrations in the other layers and the mass fraction (ratio of the mass of the layer over the mass of the particle) of layers must stay constant throughout the simulation. It seems the authors followed Couvidat and Sartelet (2015) and assumed that the transfer between the interface and the internal layer "can happen without diffusion to assure that the mass fraction of layers remain constant even during the growth or shrinking of the particle" (Page 13). This assumption may explain the unreasonable blue arrows in Figure 4 which is also questioned by the other referee. I agree with the related comments proposed by the other referee on Figure 4 and suggest revising this treatment in SOAP as the results in Figure 4 are contrary to previous kinetic simulations (please refer to Liu et al. PNAS, 2016; Mai et al. EST, 2015; Shiraiwa et al. GRL, 2012). For example, in Fig.4a, why compound **A** in the particle phase can not transfer from the core to the interface while compound **A** in the gas phase can transfer from the interface to the core? Is it just because the mass ratios of the layers have to be maintained constant as the authors stated in the last paragraph in Section 3.3?

**[ACPD](ACPD)**

Interactive
comment

**Our response**:

In the case of an extremely viscous aerosol considered here, the compounds can not transfer from the core to the interface because there is no diffusion. And the compound **A** can not transfer either from the interface to the core, but it can condense on the interface, as specified by the blue arrows.

Figure 2 (Figure 5 in the revised manuscript) theoretically presents different behaviors of volatile and low volatile organic species in a highly viscous aerosol. For these theoretical cases, mass transfer between the interface and the core is neglected because of an extremely low diffusion flux, as now specified in the paper.

Figure 2 (Figure 5 in the revised manuscript) and the corresponding text have been corrected as follows:

" Figure 5 theoretically presents different behaviors of volatile and low volatile organic species in a highly viscous aerosol. For these theoretical cases, mass transfer between the interface and the core is neglected because of an extremely low diffusion flux. The condensation of low-volatility compounds influences the behavior of higher volatility compounds.

In the case of an organic particle growth (mass increase), the condensation of a low-volatility organic compound **B** (its behavior is described by the red curved arrows in Figure 5) onto a particle can favor the condensation of a compound **A** of higher volatility (its behavior is described by blue curved arrows in Figure 5) at the interface of the particle (even if the total concentration of **A** inside the particle exceeds equilibrium). The compound **A** condenses onto the new layer created by the compound **B** to respect Raoult's law at the interface. Even though the compound **A** would evaporate if the particle was inviscid and the concentration of **A** exceeds equilibrium, for the extremely viscous case, the condensation of the compound **B** at the interface can prevent the evaporation of the compound

[Figure]

**A** stuck in the core of the particle (because of the absence of diffusion) and can lead to its "entrapment".

In the case of a shrinking particle (mass decrease), a volatile compound **A** would evaporate from the inner layers to meet the equilibrium condition if the particle is inviscid and concentration of **A** in the particle exceeds equilibrium. However, if the particle is viscous, this evaporation can strongly be slowed down, because there is no diffusion of the compound **A** from the core to the interface.

Even though the total particle mass reduces, a low-volatility compound **B** may condense at the interface and may therefore slow down the shrinking of the particle. This condensation at the interface prevents the evaporation of the compound **A** from the core of the particle.

In SOAP, a redistribution is done every time step to keep the interface thin and the mass fraction of layers constant (to prevent numerical issues: only the mass of the interface would change for a very viscous particle). This redistribution represents the fact that if the particle grows the compounds that have previously condensed are not anymore at the interface (because other compounds have condensed onto the particle) or that if the particle shrinks the compounds that were previously at the core of the particle will be eventually at the interface. Using two layers, compounds are immediately transferred between the core and the interface. A more accurate representation of the particle dynamics would be obtained using more inner layers to better represent the position of the compounds inside the particle. Nonetheless, the simulation using two layers should give a good estimation on the effect of viscosity on SOA formation. "

4. Thirdly, what is the time step adopted in the 3-D model? This is related whether the gas-particle partitioning has been reached equilibrium within the time step set in the 3-D model.

   **Our response**:

The following text has been added to the revised manuscript.

"An adaptative time step is used to solve the dynamics of organics. The minimum time step is 1 s and the maximum time step is set to 10 min in the simulations of this study. 10 min correspond to the time step used to split the different processes in the 3-D model (advection, diffusion and chemistry). When concentrations are computed by the dynamic approach, the second-order Rosenbrock scheme is used for time integration (Couvidat and Sartelet, 2015). "

5. Fourthly, the accommodation coefficient $\alpha$ is assumed as 0.5 in this study. Will the simulation results be changed if $\alpha$ is adopted as 1 (see the recent experiment study by Krechmer et al. 2017).

**Our response**:

The transition regime formula $f(Kn, \alpha)$ of Fuchs and Sutugin (1971) is used in the SOAP model.

$$f(Kn, \alpha) = \frac{0.75\alpha(1 + Kn)}{Kn^2 + Kn + 0.283Kn\alpha + 0.75\alpha} \qquad (1)$$

where $Kn$ is the Knudsen number.

The change of $\alpha$ from 0.5 to 1.0 leads to an increase in the results of the formula by about 45% compared to when $Kn = 1.0$. Therefore, the mass transfer to the particles may increase with the increase of the accommodation coefficient and the order of the increase depends on the $Kn$ value. However, other studies suggest lower accommodation coefficient (0.1, e.g., Saleh et al. (2013)).

6. Lastly, I agree with the comments proposed by the other referee that the authors stated that "the aerosols are assumed internally mixed" (Section 4.1) but the absorbing phases are still calculated separately ($c_{aq}$ and $c_p$ stated in the last paragraph in Section 4.1).

**Our response**:

The following text has been added to the revised manuscript in Section 4.1.

"The algorithm of SOAP was developped in order to consider both the organic and the aqueous phases inside a particle. It assumes that the organic and the aqueous phases co-exist in a partice but evolve separately in different regions of the particle. For example, for the dynamic representation, if a compound tends to go from the aqueous to the organic phases, it has first to evaporate to the gas phase and then condense to the organic phases instead of a direct mass transfer between the phases. It is due to the complexity of representing properly these transfers. This assumption is discussed in more details in section 2.3 of Couvidat and Sartelet (2015). "

7. I have some concern about the presentation quality. For example: the title of Section 2.1 is "Composition of aerosols" but Section 2.1 is about gas-particle partitioning.

**Our response**:

The section title has been changed in the revised manuscript as follows:

"Gas-particle partitioning for the aqueous and organic phases"

8. In Section 2.1, the authors firstly stated that in SOAP the $c_p$ included $c_{water,p}$, while in the following Eq(2) $c_{water,p}$ is calculated from $c_p$. I am confused in Eq(2) $c_p$ includes $c_{water,p}$ or not.

**Our response**:

The equation (2) for the gas-particle partitioning and the equation (3) for the water absorption are numerically solved by the Newton-Raphson method. The Newton-Raphson method is used to find better approximation to the real solution of the functions.

$$\frac{c_{p,i}}{c_{g,i}} = K_{p,i}\ c_p \tag{2}$$

$$c_{water,p}\ =\ \frac{c_p\ M_{water}\ RH}{\gamma_{water,p}\ M_p} \tag{3}$$

The schematic diagram of Figure 3 of this reply has been added to the revised manuscript for clarity.

9. Figure 2: the scenario of "SOAP-basic" is missing although it is compared with "SOAP-no_water" in the text (Page 8).

   **Our response**:

   Figure 2 (Figure 3 in the revised manuscript) has been corrected including "SOAP-basic" simulation results.

10. Figure 2 shows the diurnal variations, but the text lacks the description why the differences of simulated SOA concentrations are sometimes large in daytime while sometimes are large in nighttime.

    **Our response**:

    The following text has been added to the revised manuscript.

    " The differences between the simulations SOAP-basic and SOAP-ideal are larger during nighttime than those during daytime. This shows that the effect of ideality is larger during nighttime than daytime. This is due to the lower temperature, leading to the condensation of a larger number of organic compounds (some compounds are too volatile to condense during daytime but condense during nighttime). "

11. Page 10 Section 3.1, the authors stated that the inorganic compounds influence activity coefficients "by middle-range and long-range interactions" but as already

stated on Page 8, inorganic compounds also influence the short-range activity coefficient.

**Our response**:

The text has been corrected as follows:

" Although activity coefficients are computed with the UNIFAC model for , depending on the user's choice, in the SOAP model, activity coefficients can be calculated using the UNIFAC or the AIOMFAC model. UNIFAC was developed to reproduce the short-range interactions between water and organic compounds, which are dominant for a non-electrolyte liquid mixture. In UNIFAC, organic compounds are represented by different functional groups including alkane, aromatic carbon, alcohol, carbonyl. Interaction coefficients between water and these functional groups are calculated. However, for an electrolyte liquid mixture, the mixed organic and inorganic system may influence activity coefficients, by middle-range and long-range interactions in addition to the short-range interaction. This influence of inorganic aerosols on the calculation of activity coefficients in the SOAP model can be estimated by the AIOMFAC model that considers this mixed organic/inorganic system. "

12. Last sentence on Page 11, change "viscous compounds" to "viscous particles".

    **Our response**:

    The text has been corrected.

13. Last sentence on Page 13, "the compound **A** condenses onto the new layer created by the compound **B**" is not true. In my understanding, the layers in the model are kept same during SOA formation / evaporation.

    **Our response**:

    The sentence theoretically explains different behaviors of volatile and low volatile organic aerosols in a highly viscous aerosol.

The text has been corrected as follows:

" In the case of an organic particle growth (mass increase), the condensation of a low-volatility organic compound **B** (its behavior is described by the red curved arrows in Figure 5) onto a particle can favor the condensation of a compound **A** of higher volatility (its behavior is described by blue curved arrows in Figure 5) at the interface of the particle (even if the total concentration of **A** inside the particle exceeds equilibrium). The compound **A** condenses onto the new layer created by the compound **B** to respect Raoult's law at the interface. "

14. Section 4.2, besides MFE and MFB, RMSE should also be defined.

    **Our response**:

    The text has been corrected.

15. OM1 etc in Table 4 should also be defined.

    **Our response**:

    Definitions are added to Table 4 as follows:

    " subscripts are used for the particle size. For example, $OC_{2.5}$ is organic carbon of aerodynamic diameter lower than 2.5 m. For ammonium ($NH_4$), sulfate ($NH_4$) and nitrate ($NO_3$), e.g., $SO_{4,1}$ is sulfate of aerodynamic diameter lower than 1 m. "

16. Section 4.3, it is better to describe Fig 6 from Fig 6a to Fig 6d instead of beginning from Fig 6d.

    **Our response**:

    The text has been corrected describing from Fig 6a to 6d in the revised manuscript.

17. The first sentence in Section 4.4, "influence" should be "influenced".

**Our response**:

The text has been corrected.

References:

- Krechmer et al. 2017, Direct Measurements of Gas/Particle Partitioning and Mass Accommodation Coefficients in Environmental Chambers, Environmental Science & Technology, 51(20), 11867-11875.

- Liu, P., Y. J. Li, Y. Wang, M. K. Gilles, R. A. Zaveri, A. K. Bertram, and S. T. Martin (2016), Lability of secondary organic particulate matter, Proceedings of the National Academy of Sciences, 113(45), 12643-12648.

- Mai, H., M. Shiraiwa, R. C. Flagan, and J. H. Seinfeld (2015), Under What Conditions Can Equilibrium Gas–Particle Partitioning Be Expected to Hold in the Atmosphere?, Environmental Science & Technology, 49(19), 11485-11491.

- Shiraiwa, M., and J. H. Seinfeld (2012), Equilibration timescale of atmospheric secondary organic aerosol partitioning, Geophysical Research Letters, 39(24).

**2   Reply to anonymous referee #2's comments**

**General comments**

This manuscript addresses the presentation of secondary organic aerosols in air quality model, specifically the non-ideality and viscosity of the particles. This is an important topic on the field of SOA studies and the topic is well suited for ACP. I find some points that should be addressed before publishing in ACP.

**Specific comments**

1. Authors present organics with few surrogate compounds and this is understandable approach in 3D air quality model. There is no discussion in the manuscript about the effect of choice of the surrogate compounds on the results and I wonder if the authors could comment this little bit. First, how does the use of 21 surrogate compounds (instead of the huge number of organics in the real atmosphere) affect the results regarding the non-ideality and the activity coefficient calculations; Is it expected that activity coefficients matter that much also when the particles consist of mixtures of much more compounds?

**Our response**:

The text has been modified in the revised manuscript as follows:

" The effect of activity coefficients was already investigated in a previous study (Couvidat et al., 2012) by using the UNIFAC model. Compared to assuming ideality, computing activity coefficients was found to decrease the concentrations of hydrophobic SOA (condensing onto the organic phase of particles) but also to increase the concentrations of hydrophilic SOA (condensing onto the aqueous phase of particles). AIOMFAC and UNIFAC are used in this study to compute the activity coefficients for organic-inorganic mixture. These models have been developed using a group contribution method. 18 main functional groups and 45 subgroups in AIOMFAC are used in this study. The SOA surrogates are splitted into these functional groups. The computation of activity coefficients depends on the functional groups that are present in the SOA surrogates. It is assumed here that the SOA surrogates represent the major SOA compound types in terms of functional groups. However, considering more compounds in the model may affect the computation of activity coefficients, and enhance their effect as a stronger variability of composition would be simulated. "

2. Second, how does the choice of surrogate compounds affect the viscosity effect? For instance, monoterpene oxidation products seem to be missing low-volatility compounds (unless there is acidic aqueous phase). If the mass of low-volatility

compounds was underestimated, wouldn't that lead to viscosity effect being over-estimated?

**Our response**:

The following text has been added to the revised manuscript at the end of Section 4.5.

" The viscosity effect is very low for low-volatile compounds (Couvidat and Sartelet, 2015). Here, extremely low volatile compounds from the oxidation of monoterpenes are not modeled (Chrit et al., 2017). Taking them into account may decrease the viscosity effect estimated in this study. "

3. There are some aspects in the description of dynamic calculations for viscous particles where some clarification would be helpful: 1) Eq. 7 seems to describe the diffusion in particle phase based on difference between equilibrium particle phase concentration and the actual particle phase concentration of compound i. Based on the given reference (Couvidat and Sartelet 2015), in eq. 7 the concentrations should be for each layer, not the total particle phase concentrations, right?

**Our response**:

The concentrations in Equation 7 represent the concentrations for each particle layer. The text has been modified as follows:

" This deviation can be described by taking into account the flux of diffusion with the mass transfer rate by condensation/evaporation for each particle layer (Equation 36 of Couvidat and Sartelet (2015)).

$$J_{diff}^{layer} = k_{diff}^{layer} \left( c_{g,i} K_{p,i}^{layer} c_p^{layer} - c_{p,i}^{layer} \right) \tag{7}$$

"

4. 2) The equation in the given reference seems to be derived for a case where the particle phase concentration at the interface is constant (equilibrium with gas phase). This is not the case here if also condensation/evaporation is calculated dynamically. I wonder if the equation is correct to use here.

**Our response**:

The concentration at the interface is not constant in the dynamic approach used in this study. In the simplified method, it is assumed that the concentrations in one layer can be described independently from the concentrations in the other layers, but they depend on the diffusion. When the evolution of $c_{p,i}^{layer}$ in a bin and a layer is limited by the diffusion, the deviation in the evolution of $c_{p,i}^{layer}$ is estimated compared to the equilibrium concentration ($c_{g,i}K_{p,i}^{layer}c_p^{layer}$). This model has been compared to an explicit model. For an extremely viscous aerosol, we have obtained very similar evolution of $c_{p,i}^{layer}$ using our simplified method and the explicit method.

5. 3) If the $J_{diff}$ in eq. 5 is calculated for each layer as in the reference, then it is not clear which $J_{diff}$ is used in eq. 10. for calculating $J_{tot}$.

**Our response**:

$J_{diff}$ in Equation 10 (Equation 11 in the revised manuscript) is the sum of the diffusion fluxes over all aerosol layers. The diffusion flux for each layer is now presented as $J_{diff}^{layer}$ in Equations 7 and 10 of the revised manuscript.

" We assume that in each particle layer the evolution of concentration $c_{p,i}^{layer}$ of species $i$ can be described as a deviation of an equilibrium concentration ($c_{g,i}K_{p,i}^{layer}c_p^{layer}$) when the condensation/evaporation of the species is limited by the diffusion of organic compounds in the organic phase.

This deviation can be described by taking into account the flux of diffusion with the mass transfer rate by condensation/evaporation for each particle layer (Equation

36 of Couvidat and Sartelet (2015)).

$$J_{diff}^{layer} = k_{diff}^{layer} \left( c_{g,i} K_{p,i}^{layer} c_p^{layer} - c_{p,i}^{layer} \right) \quad (7)$$

where the concentrations $c_p^{layer}$ and $c_{p,i}^{layer}$ are the values for each particle layer.

The kinetic rate of diffusion $k_{diff}^{layer}$ $(s^{-1})$ is computed as follows (Couvidat and Sartelet, 2015):

$$k_{diff}^{layer} \propto \frac{1}{\tau_{diff}} \quad (8)$$

$\tau_{diff}$ is the characteristic time (s) for diffusion in the particle:

$$\tau_{diff} = \frac{R_p^2}{\pi^2 D_{org}} \quad (9)$$

where $R_p$ is the radius of the particle (m) and $D_{org}$ is the organic-phase diffusivity (m$^2$/s).

The sum of the diffusion fluxes over all aerosol layers is obtained as follows:

$$J_{diff} = \sum_{layer} J_{diff}^{layer} \quad (10)$$

The final mass flux by the mixed phenomenon condensation/evaporation/diffusion for the particle is computed by assuming that the characteristic time of the combined effect of condensation/evaporation and diffusion is equal to the sum of the characteristic time of condensation/evaporation ($J_{cond/evap}$) and the sum of the diffusion fluxes over all aerosol layers ($J_{diff}$) as follows:

$$\frac{1}{J_{tot}} = \frac{1}{J_{cond/evap}} + \frac{1}{J_{diff}} \quad (11)$$

"

6. 4) Only two layers are used for calculations here. Is the model accurate with only two layers? In the given reference the model is evaluated only for three or more layers.

**Our response**:

Using two layers, compounds are immediately transferred between the core and the interface. A better representation of the particle dynamics would probably be obtained using more inner layers by better representing the position of compounds inside the particle. Nonetheless, the simulation using two layers should give a good estimation on the effect of viscosity on SOA formation.

As the viscosity is assumed to be too high for diffusion to be significant, a great number of layers are not needed as long as the condensation and evaporation of compounds at the interface (that will drive the evolution of the particle) is well represented.

Table 5 shows the computation time for the Dynamic Viscous simulation is ten times than that for the Equilibrium UNIFAC simulation. This increase in computation time is very important for 3-D air quality modeling. Simulations with more layers and more realistic values of viscosity would need to be conducted with more computer resources in the futur.

7. It is stated (P14, L20) that the aerosols are assumed internally mixed, but still different absorbing masses are used for hydrophilic and hydrophobic compounds. Are the two phases assumed to co-exist in a particle? How is this taken in to account in the dynamic approach where equations for spherically symmetrical cases are used (eq. 5-9)?

**Our response**:

We assume that the organic and the aqueous phases co-exist in a partice but evolve separately in the SOAP model. For example, if a compound tends to go from the aqueous phase to the organic phase, it has first to evaporate to the gas

phase and then condense to the organic phase instead of a direct mass transfer between the phases. It is due to the compelxity of representing properly these transfers. This assumption is discussed in more details in section 2.3 of Couvidat and Sartelet (2015).

8. Figure 4 "Shrinking" part and related discussion on P14 L6-9: It is not clear how shrinking in this conceptual discussion is thought to happen. Could the authors clarify this? Especially confusing is the blue text about volatile compound A being stuck at the interface due to shrinking. In case of a net mass flux of A towards the particle, A could be stuck at the interface and not transferred to the particle core. But how does that happen at the same time when A is evaporating and the particle thereby shrinking?

**Our response**:

In the SOAP model, a particle is represented by a mixture of many different species. Even though the particle is "shrinking", i.e., a decrease of total particle mass, some species may condense on the particle to meet the equilibrium condition.

For the corrections of the text and Figure 4, please see our answer to no. 3 of the reviewer 1.

**Technical comments**

1. Table 1, row BiA0D, last column: A typo: "qaueous".

   **Our response**:

   The text has been corrected.

2. Table 1: Units missing for Kp.

   **Our response**:

The missing unit has been added.

3. P8, L15: Text refers to "SOAP-basic" although such simulations is not presented in the fig. 2.

    **Our response**:

    Figure 2 (Figure 3 in the revised manuscript) has been corrected.

4. P10, L18: "concentrations" twice.

    **Our response**:

    The text has been corrected.

**References**

Abramson, E., Imre, D., Beranek, J., Wilson, J., and Zelenyuk, A.: Experimental determination of chemical diffusion within secondary organic aerosol particles, Phys. Chem. Chem. Phys., 15, 2983–2991, doi:10.1039/C2CP44013J, 2013.

Choi, M. Y. and Chan, C. K.: The effects of organic species on the hygroscopic behaviors of inorganic aerosols, Environ. Sci. Technol., 36, 2422–2428, doi:10.1021/es0113293, 2002.

Chrit, M., Sartelet, K., Sciare, J., Pey, J., Marchand, N., Couvidat, F., Sellegri, K., and Beekmann, M.: Modelling organic aerosol concentrations and properties during ChArMEx summer campaigns of 2012 and 2013 in the western Mediterranean region, Atmos. Chem. Phys., 17, 12 509–12 531, doi:10.5194/acp-17-12509-2017, 2017.

Couvidat, F. and Sartelet, K.: The Secondary Organic Aerosol Processor (SOAP v1.0) model: a unified model with different ranges of complexity based on the molecular surrogate approach, Geosci. Model Dev., 8, 1111–1138, doi:10.5194/gmd-8-1111-2015, 2015.

Couvidat, F., Debry, É., Sartelet, K., and Seigneur, C.: A hydrophilic/hydrophobic organic ($H^2O$) model: Model development, evaluation and sensitivity analysis, J. Geophys. Res., 117, D10 304, doi:10.1029/2011JD017214, 2012.

DeRieux, W.-S. W., Li, Y., Lin, P., Laskin, J., Laskin, A., Bertram, A. K., Nizkorodov, S. A., and Shiraiwa, M.: Predicting the glass transition temperature and viscosity of secondary

organic material using molecular composition, Atmos. Chem. Phys., 18, 6331–6351, doi: 10.5194/acp-18-6331-2018, 2018.

Fuchs, N. and Sutugin, A.: High-dispersed aerosols, in: Topics in current aerosol research, edited by Brock, G. H., International reviews in aerosol physics and chemistry, Pergamon, http://www.sciencedirect.com/science/article/pii/B9780080166742500066, 1971.

Maples, R. E.: Petroleum Refinery Process Economics, Pennwell Books, 2nd ed., 2000.

Pfrang, C., Shiraiwa, M., and Pöschl, U.: Chemical ageing and transformation of diffusivity in semi-solid multi-component organic aerosol particles, Atmos. Chem. Phys., 11, 7343–7354, doi:10.5194/acp-11-7343-2011, 2011.

Saleh, R., Donahue, N. M., and Robinson, A. L.: Time scales for gas-particle partitioning equilibration of secondary organic aerosol formed from alpha-pinene ozonolysis, Environ. Sci. Technol., 47, 5588–5594, doi:10.1021/es400078d, 2013.

Song, Y. C., Haddrell, A. E., Bzdek, B. R., Reid, J. P., Bannan, T., Topping, D. O., Percival, C., and Cai, C.: Measurements and predictions of binary component aerosol particle viscosity, J. Phys. Chem. A, 120, 8123–8137, doi:10.1021/acs.jpca.6b07835, 2016.

**Fig. 1.** Concentration of total condensed water in the SOAP-Reference simulation ($\mu$g m-3).

A volatile
B low volatile

Growth

**A** cannot go to the
interface (and possibly
evaporate) because of the
absence of diffusion

**A** condenses to
respect
Raoult's law at
the interface
(even if total
particle
concentrations
are above
equilibrium)

**B** does not need
a lot of mass to
condense

Shrinking

The evaporation
of **A** is reduced
because of the
absence of
diffusion and **A**
is stuck at the
interface due to
the shrinking
(even if total
particle
concentrations
are below
equilibrium)

**B** does not need
a lot of mass to
condense

**Interface
richer in B**

**Fig. 2.** Schematic representation of SOA formation for a growing (top) and shrinking (down)
highly viscous aerosol (Blue arrows: an organic compound A and red arrow: a low-volatility
organic compound B).

[Figure]

**Fig. 3.** Computation steps for the gas-particle partitioning and the water absorption.